# SUMO and Transcriptional Regulation: The Lessons of Large-Scale Proteomic, Modifomic and Genomic Studies

**DOI:** 10.3390/molecules26040828

**Published:** 2021-02-05

**Authors:** Mathias Boulanger, Mehuli Chakraborty, Denis Tempé, Marc Piechaczyk, Guillaume Bossis

**Affiliations:** 1Institut de Génétique Moléculaire de Montpellier (IGMM), University of Montpellier, CNRS, Montpellier, France; mathias.boulanger@embl.de (M.B.); mehuli.chakraborty@igmm.cnrs.fr (M.C.); denis.tempe@igmm.cnrs.fr (D.T.); 2Equipe Labellisée Ligue Contre le Cancer, Paris, France

**Keywords:** SUMO, transcription, transcription factors, transcriptional coregulators, chromatin, chromatin modification, heterochromatin

## Abstract

One major role of the eukaryotic peptidic post-translational modifier SUMO in the cell is transcriptional control. This occurs via modification of virtually all classes of transcriptional actors, which include transcription factors, transcriptional coregulators, diverse chromatin components, as well as Pol I-, Pol II- and Pol III transcriptional machineries and their regulators. For many years, the role of SUMOylation has essentially been studied on individual proteins, or small groups of proteins, principally dealing with Pol II-mediated transcription. This provided only a fragmentary view of how SUMOylation controls transcription. The recent advent of large-scale proteomic, modifomic and genomic studies has however considerably refined our perception of the part played by SUMO in gene expression control. We review here these developments and the new concepts they are at the origin of, together with the limitations of our knowledge. How they illuminate the SUMO-dependent transcriptional mechanisms that have been characterized thus far and how they impact our view of SUMO-dependent chromatin organization are also considered.

## 1. Introduction

Since the beginning of the 2000s, there has been accumulating evidence that the eukaryotic peptidic post-translational modifiers of the Small Ubiquitin-like Modifier (SUMO) family (collectively termed SUMO hereafter) plays an important and multifaceted part in the regulation of gene transcription. This is achieved via their conjugation to transcription factors (TFs), positive and negative transcriptional coregulators, histones and other chromatin proteins, chromatin-remodeling complexes, chromatin-modifying enzymatic systems, as well as to the transcriptional machineries themselves. Indeed, one of the first roles that has been characterized for SUMO is regulation of transcription. It is also the most documented one in the literature [1,2,3,4,5].

Although this review principally deals with RNA polymerase II (Pol II)-mediated transcription regulation by SUMO in mammalian cells, major concepts originating from other organisms will also be addressed together with transcription by RNA Polymerases I (Pol I) and -III (Pol III). 

## 2. SUMOylation: A Principally Nuclear Post-Translational Modification

A number of preliminary considerations on SUMOylation itself are required before getting into the mechanistic details of gene regulation by SUMO. For more general information on protein SUMOylation and its biological consequences, the reader is referred to general reviews [6,7,8,9,10,11]. Of note, the recent one by Celen and Sahin [10] provides an interesting historical perspective of the field since the discovery of SUMO 25 years ago and that by Chang and Yeh [11] a focus on SUMO in diseases. 

### 2.1. SUMO Isoforms and the General Effects of SUMOylation on Proteins

The term SUMO defines a group of polypeptides of approximately 100 amino acids belonging to a broader family of eukaryote-specific peptidic post-translational modifiers called the ubiquitin-likes (Ubls). Their founder, ubiquitin, is a major actor in, not only proteasomal degradation of intracellular proteins, but also many non-proteolytic signaling events [12,13,14]. 

SUMO (initially called sentrin) is among the most abundant cellular proteins with, for example, approximately 10^7^ molecules/cell in the human cervix cancer cell line HeLa, i.e., only slightly less than its kin ubiquitin [15]. Like ubiquitin and the other Ubls, SUMO is dynamically and reversibly conjugated to lysines of a myriad of protein substrates, of which it modifies the fate and/or function [6,7,9,10]. This can be achieved by inducing internal conformational changes that may change intrinsic properties of the concerned substrates, including changes of affinity for binding partners. SUMOylation can also hide or reveal protein-protein- or protein-nucleic acid interaction domains, provide the ability to interact with SUMO-binding proteins, affect subcellular distribution, compete with other PTMs and/or alter enzymatic activity or protein turnover [6,7,9,10]. Importantly, SUMOylation can occur, not only on isolated proteins, but also in supramolecular complexes where its molecular effects are nevertheless often more difficult to decipher (see group SUMOylation section below). Additionally, at least under conditions of acute stress where it actively participates in the cell adaptive response, SUMO has also been proposed to increase the solubility of certain of its substrates to avoid the formation of toxic proteinacious aggregates [16,17]. SUMOylation can also favor protein solubility via inhibiting the formation of paracrystals by a transcription factors such as STAT1 and -3 [18]. All these mechanisms may affect transcription regulation in a way or another and are not necessarily mutually exclusive.

SUMO is present in different forms and numbers in the various eukaryotes, ranging from one in an organism such as yeast up to eight in certain plants [19,20]. In mammals, it consists of a well-characterized family of three ubiquitous members (SUMO-1 to -3; collectively called SUMO herein), which have received considerable attention, plus two other paralogs (SUMO-4 and -5) with tissue-restricted expression and whose functions are still unclear [10,11]. The latter two SUMOs will therefore not be considered here. Importantly, SUMO-2 and -3 are approximately 50% identical to SUMO-1 and 97% identical between them. Paralog specificity or preference between SUMO-1, on one side, and SUMO-2 and -3, on the other, have been described in several occurrences [6,7,9,10,11], as well as in large-scale proteomic approaches [21]. In contrast, the differences between SUMO-2 and SUMO-3 are much less documented, as, due to their high sequence conservation, tools allowing to discriminate them functionally and/or biochemically in vivo are currently limiting. They have therefore often been studied together and, in this case, named SUMO-2/3. Paralog preference may also, in part, be accounted for by differences in the relative abundances of the different SUMOs. Thus, although the 3 SUMOs are mostly found conjugated to their protein substrates in living cells, free SUMO-1 is much less abundant than free SUMO-3, which is itself usually less abundant than SUMO-2. Moreover, even though the free SUMOs are most often assumed to diffuse evenly within the cell due to their small size, there also exist occasional indications that their distribution might be dynamically affected by signaling and/or stresses (see ref. [22] for an example).

### 2.2. The SUMOylation Cycle

Several sets of enzymes are involved in the SUMOylation cycle (Figure 1). First, the SUMO polypeptides are synthetized in the form of precursors. They are proteolytically matured by members (SENP1, -2, -3 and -5) of the SENP (SENtrin-specific proteases) family of proteases (also comprising SENP6 and -7; see below) which cleave their (short) C-terminal extension [23]. This exposes a diglycine C-terminal motif essential for covalent binding to substrates. This occurs in several steps and ends by the formation of an isopeptide bond between the C-terminal carboxyl group of processed SUMO and the ε-amino group of lysines targeted on substrates. First, processed SUMOs are transferred in an ATP-dependent manner to a unique cellular SUMO-activating enzyme (SUMO E1) formed of two subunits, SAE1 (also called Aos1) and SAE2 (also called Uba2). This is achieved via a two-step mechanism leading to the formation of a high energy thioester bond between the carboxyl group of the SUMO C-terminal glycine and a reactive cysteine (C173) located in the catalytic core borne by the SAE2 subunit. Activated SUMO is then transferred to a unique cellular SUMO-conjugating enzyme (SUMO E2 or Ubc9), again via a thioester bond. Finally, the SUMO is covalently attached to target substrates from the E2 enzyme with, in many cases, the help of members of a family of SUMO E3 factors guiding the targeted lysine of substrates to the active site of the E2 enzyme.

The SUMO E3s characterized so far are relatively few in number. The best-studied ones include the members of the PIAS family (PIAS-1 to -4; with spliced variants for certain of them), several TRIM family members, RANBP2 or the Pc2/CBX4 component of the Polycomb complex [10,11,24]. Certain histone deacetylases that have roles in transcription regulation have also been proposed to display SUMOylation activity [25,26,27]. This is also the case of at least one transcription factor, hDREF, which facilitates the SUMOylation of the chromatin remodeler Mi2α [28] (also see below). Importantly, at least one SUMO E3, ZNF451 [29], also displays an E4 elongase activity promoting the formation of SUMO chains essential for certain signaling events (see below). Besides this, certain SUMO E3s, including several PIAS proteins, can also exert a role in transcription independently of their SUMOylation activity (see below).

Finally, SUMOylated proteins can be deSUMOylated and SUMO chains can be depolymerized by SUMO isopeptidases (also known as deSUMOylases) [23,30,31]. The deSUMOylases identified so far are SENP-1 to -3, SENP-5 to -7, DeSI-1 and -2 and USPL1. Noteworthy SENP-1 shows preference for SUMO-1 whereas the other SENPs preferentially remove SUMO-2/3 [23,30,31], but substrate preference displayed by the different SUMO isopeptidases is partly explained by their differential subcellular distribution [23]. Also of note, SENP6 and -7 are preferentially involved in SUMO chain disassembly [30,31]. Importantly, SUMO is a quite stable protein and can be recycled for further protein modifications.

There is now ample evidence that protein SUMOylation can no longer be considered in an isolated manner due to multiple cross-talks with protein modification by other Ubls. For example, the FAT10 Ubl can inhibit the activation of the SUMO-1, -2 and -3 via covalent binding to the catalytic cysteine of the SUMO E1 [32]. Yet, it cannot be transferred to another protein by Fat10-loaded SUMO E1 [32]. Moreover, ubiquitin and certain Ubls, including SUMO, can form mixed chains, sometimes of complex and still ill-defined composition and topologies and can also be subjected to other post-translational modifications as discussed below. Elucidating the biochemical complexity and the biological consequences of what is now called the “ubiquitin code” is currently the object of intensive studies and still requires considerable efforts [33,34,35,36]. This implies that a number of conclusions from studies carried out before the advent of large-scale proteomics/modifomics might have to be reconsidered or, at least, nuanced. 

### 2.3. SUMO Substrates and Nuclear Functions of SUMOylation

Thus far, proteomic approaches have characterized >6700 proteins SUMOylated at >40,000 sites, usually in flexible and disordered peptidic domains [37,38]. It is, however, possible that the actual number of SUMOylated proteins is higher due to the limitations of the current technologies and the still limited number of studies. These figures are coherent with the wide variety of cellular processes, the regulation of which was shown to be impacted by SUMO in numerous functional studies [6,7,9,10,11]. Moreover, SUMOylated proteins are essentially found in the cell nucleus. Thus, >80% of the cellular SUMOylated proteins characterized in proteomic approaches were identified as nuclear with >70% of nuclear proteins seen to be SUMO substrates at a point or another [37,38]. Consistently, SUMO has principally been associated with nuclear functions so far, especially at chromatin, as well as in several other nuclear substructures. The characterized nuclear functions of SUMO include, nucleo-cytoplasmic exchanges [39], the various responses to DNA damages [40,41], mRNA processing and metabolism [42], chromosome biology and chromatin organization [24,31,43] and gene transcription regulation [3,4,24,31,44]. There also exist SUMO-dependent cross-talks between these functions, which are often concurrent, coordinated and sometimes intermingled. In the context of this review, a first obvious cross-talk to mention is that between transcription and maturation of RNAs. But there may also exist others between nuclear functions controlled by SUMO. This is, for example, (i) the case of transcription-coupled nucleotide excision repair which involves both ubiquitin and SUMO for its resolution to correct UV-generated DNA mutations [45], (ii) the resolution of conflicts between transcription and DNA replication [46] and (iii) intranuclear positioning of genes for optimal transcription [47]. In addition to chromatin (see below), the nuclear structures implicating the SUMO pathway include (i) Cajal bodies [42], which are involved in the biogenesis of snRNPs and snoRNPs, as well in maintenance of telomeres and processing of histone mRNAs, (ii) the nuclear splicing speckles [42], which are involved in RNA polymerase II-transcribed RNA maturation, (iii) the nucleolus [2,48], which is involved in the biogenesis of ribosomes as well as the maturation of the U6 and U6atac snRNAs, (iv) PcG bodies, which are sites of transcriptional silencing [49,50], (v) histone locus bodies, which are sites of production and maturation of histone mRNAs [51] and (vi) PML bodies, which are multifunctional structures enriched in SUMOylated proteins that are capable of regulating transcription [52]. 

Many proteomics studies have now addressed the precise nature of both SUMOylated proteins and SUMOylated sites at the system-wide level in different species (including mammals, drosophila, worms, plants and yeast), sometimes in pathological situations [17,21,53,54,55,56,57,58,59,60,61,62,63,64,65,66,67,68,69,70,71,72,73,74,75,76,77,78,79,80,81,82,83,84,85,86,87,88,89]. One of their general conclusions is that SUMOylated proteins are largely functionally connected. This supports the notion developed below that proteins can be SUMO-modified in group according to molecular mechanisms that largely remain to be clarified. Thus, besides pre-mRNA splicing, ribosome and ribosome biogenesis, DNA damage repair and cell cycle regulation, transcriptional regulation and chromatin remodeling appeared to constitute two of the main functional clusters identified in these studies [37]. Through the years, these proteome-wide studies have also been preceded or complemented by more focused studies on specific proteins, or group of proteins, playing a part in transcription. Non exclusively, these proteins include: (i) transcription factors [3,4,90], (ii) negative transcriptional coregulators such as certain histone acetyl transferases (HDACs) or lysine-specific demethylases (KDMs), (iii) positive transcriptional regulators such as CBP/p300, (iv) various components of chromatin-remodeling complexes and -regulators, (v) components of the Pol-I, -II and -III transcription machineries and (vi) histones [37,38]. Their respective cases will be considered later in this review.

### 2.4. Cross-Talks and Competition of SUMOylation with Other Post-Translational Modifications: Possible Consequences for Transcription

In the regard of its predominant conjugation to nuclear proteins, SUMOylation departs significantly from other post-translational modifications (PTMs) such as phosphorylation, acetylation or ubiquitylation, which largely occur throughout the whole cell. This however does not prevent SUMO to crosstalk with other PTMs. Thus, SUMO substrates can also be modified by acetylation, methylation or ubiquitylation or other PTMs occurring on lysines (see below) but also by other PTMs such as phosphorylation occurring on other amino acid residues. Moreover, the SUMO proteins themselves an also be subjected to PTMs [38]. Taken with the fact that the three SUMOs can form chains, which can be branched and which can also involve other members of the ubiquitin family [10,30,31,37], these observations point to a long underappreciated complexity of the SUMO signal itself. A considerable amount of work is still needed to fully characterize it biochemically and functionally, including for better understanding of transcriptional control in cells.

Another striking difference with many other PTMs is that the fraction of a given protein that is SUMOylated at a given time in the cell is very low (usually below, or much below, the percent), at the exclusion of a few exceptions such as the RanGAP1 nucleopore-associated protein or the PML component of the PML nuclear bodies [6,7,9,10]. Such an observation holds for all transcriptional actors, the SUMOylation of which has been investigated so far [3]. This undoubtfully has mechanistical reasons that have rarely been studied experimentally in detail up to now. A first possibility is that transient modification of a protein, which is rendered possible by high deSUMOylase activity in the cell, might be sufficient for irreversible transition from one functional state to another. Another possibility is that SUMOylation may concern only the small fraction of a protein (possibly topologically confined within the cell) that is sufficient to permit a given biological response. Finally, neither all proteins nor all SUMOylation sites may need to be SUMOylated in certain supramolecular complexes, as discussed below when addressing the notion of group SUMOylation. It is obvious that these non-exclusive possibilities may apply to transcriptional control, especially when taking into account notions as simple and well-established as (i) not all molecules of a given TF are active at a given time in the cell and (ii) TFs do not work in an isolated manner but in interaction with numerous chromatin factors and sometimes in relatively stable complexes [91,92].

Another paramount point to consider is that numerous lysines identified as SUMOylated in vivo can also be ubiquitylated, acetylated or methylated as already mentioned [37]. On its own, this is indicative of possible competitions between these PTMs. The first one to be documented was that between ubiquitylation and SUMOylation of the IκBα inhibitor of the TF NFκB [93]. Competition between SUMOylation and ubiquitylation may however not necessarily occur on the same lysine residues, as may be the case in the TFs Oct4 [94] and ATF3 [95]. Competition between SUMOylation and acetylation of the same lysines of different transcription factors have also been reported. This is, for example, the case of the tumor suppressor HIC1 [96] and of the intracellular form of lactoferrin, delta-lactoferrin [97]. This translates into differential transactivation activity of the two TFs depending on their post-transcriptional status, i.e., negative when SUMOylated and positive when acetylated. This also holds true for the KLF8 oncoprotein [98], as well as TFs STAT5 [99], MEF2 [25,100] and DAXX [101], acetylation altering their SUMO-dependent protein interaction networks by preventing binding of SUMO-interacting co-factors. Such a competition between SUMOylation, on one side, and ubiquitylation, acetylation or methylation, on the other, must seriously be borne in mind, as the latter PTMs have extensively been associated with gene expression- and chromatin biology regulation [102,103,104,105]. Additionally, lysines are also subjected to a variety of other less well characterized PTMs, including non-enzymatic ones [106]. At this stage of investigation, it cannot be excluded that they could also compete with SUMOylation, at least on certain substrates and possibly during certain transcriptional events. This is all the more to be considered that histones and other chromatin proteins, including transcriptional actors, are known to be subjected to short chain acylations other than acetylation. These include propionylation, butyrylation, 2-hydroxyisobutyrylation, succinylation, malonylation, glutarylation, crotonylation and β-hydroxybutyrylation [107].

The interplay between SUMOylation and other PTMs can also occur on different amino acid residues. This is, for example, the case between SUMOylation and phosphorylation at closely located lysines and threonines residues in the TFs STAT1 [18] and STAT5 [99] where they are antagonistic. Such an interplay can also occur on a longer distance. This concerns the TF Fos where phosphorylation of a Ras-responsive threonine (T232) can inhibit SUMOylation at a distant lysine (K265) [108], as well as in its transcriptional partner Jun where phosphorylation by the kinase JUNK in its N-terminal domain inhibits SUMOylation in the C-terminal moiety of the protein [109]. Along the same line, acetylation and SUMOylation at distant lysines of the nuclear farnesoid X receptor (FXR) were reported to be antagonistic as well, the former favoring transcription of inflammatory genes in liver and the latter the opposite [110]. In contrast, an acetylation-SUMOylation switch whereby acetylation of the nuclear receptor pregnane X receptor (PXR) constitutes a prerequisite for its subsequent SUMOylation has been characterized and shown important for repression of PXR-target gene expression [111]. 

Importantly, interactions between PTMs are not necessarily antagonistic but can also be collaborative at the substrate level. For example, certain phosphorylations are known to promote SUMOylation of substrates at so-called phosphorylation-dependent extended consensus SUMOylation motifs (PDSMs; ΨKxExxSP where Ψ is a large hydrophobic amino acid and x any amino acid) found in many (but not only) TFs [6,7,10,37,112]. Notably, a recent proteomic survey showed that 9% of SUMOylation events occur proximal to phosphorylated amino acids, a fraction of them being dependent on these phosphorylations, including in chromatin proteins and transcriptional regulators [38] such as the TF Bcl11b [113]. Also of note, antagonisms or collaborations between PTMs can involve more than 2 of them on the same substrate. For example, the acetyl/SUMO switch described above for the TF MEF2 also involves phosphorylation/dephosphorylation events [114].

### 2.5. The Specific Case of SUMO Crosstalk with Ubiquitin: when SUMO Chains Come into Play

The diversity of protein modifications by SUMO itself and the crosstalk with ubiquitylation are not necessarily mutually exclusive and may have possible consequences on transcriptional control. Not only proteins can be monoSUMOylated, possibly at multiple sites (multiSUMOylation), but also polySUMOylated (possibly at multiple sites; multipolySUMOylation), i.e., subjected to modification by SUMO chains that can be of different natures [30,31]. Thus, whereas SUMO-1 cannot form homomeric chains, SUMO-2 and -3 can. This occurs preferably on their lysine 11 (not conserved in SUMO-1), which is part of a SUMOylation consensus site, but can also take place at alternative non-consensus SUMOylation sites [73]. However, SUMO-1 can be used to cap and terminate SUMO-2/3 chains. The functional consequences of these different types of SUMOylation can be different, even though their biochemical consequences are largely based on a similar mechanism, i.e., the ability of SUMO to interact via relatively weak non-covalent interactions with other polypeptides bearing so-called SUMO-interacting motifs (SIMs). SIMs are most often, but not always, short hydrophobic amino acids motifs (V/I-x-V/I-V/I) followed by acidic- (D, E) or phosphorylatable (S, T) amino acid stretches. SUMO-SIM interactions can promote the interaction of SUMOylated proteins with other polypeptides [6,7,9,10,11]. Interactions between SUMO and other proteins is however not limited to short hydrophobic SIM recognition, as at least two other types of SIMs binding to distinct surfaces of SUMO have been described [115,116,117]. MonoSUMOylation is usually considered to be mostly involved in the assembly of protein complexes or to have intramolecular effects. In contrast, the outcomes of multimono- or polySUMOylation also include the triggering of protein destruction, as well as diverse non-proteolytic processes. In the latter case, polySUMO chains are disassembled by SENP6 and -7 SENP family members after termination of the signaling events they have been involved in [30,31]. An interesting discovery of the past years is that multimonoSUMOylated- or polySUMOylated proteins can recruit multiple SIMs-containing proteins called SUMO-targeted Ubiquitin E3 ligases (or StUbLs). This is followed by ubiquitylation of SUMO substrates and/or SUMO itself [30,31], which can lead to either proteasomal proteolytic destruction or another destiny, as recently shown for centromeric proteins [118] and certain transcription factors (see hereafter).

RNF4 was the first StUbL identified in mammals [119,120,121]. It contains 4 N-terminally located SIMs showing preference for SUMO-2 and a C-terminal ring finger domain essential for ubiquitylation. An unbiased proteomic study conducted in human U2OS cells has shown that RNF4 targets a large number of proteins for proteasomal destruction. If the main category of substrates is the SUMOylation machinery itself, many of the other substrates deal with nucleic acids biology. The main processes characterized include DNA repair, which is consistent with the documented role of RNF4 in this process [122], but also transcription, by affecting factors interacting with, or contributing to, the TFIID complex and the Pol II machinery [121]. Transcription factors such as Sp1 [123] or Myc [124] are also concerned. However, the effect seems context-dependent in the latter case, as SUMOylation of Myc can also lead to protein stabilization during lymphomagenesis [125]. It ensues from this that part of the homeostasis of essential cell transcriptional actors depends on the balance between the activity of RNF4, which stimulates the breakdown of these factors via SUMO-dependent ubiquitylation, and that of SENP6 and/or -7, which trim SUMO chains.

Arkadia (also called RNF111) is the second mammalian StUbL identified so far. It harbors 3 SIMs that preferentially recognize SUMO-2/3 chains capped by SUMO-1 over homotypic SUMO-2/3 chains [126]. If it is well established that it plays a ubiquitin-dependent non-proteolytic part in DNA repair [127], it also has been shown to regulate the activity of several transcription factors following both proteolytic and non-proteolytic mechanisms. The actual role of SUMOylation in Arkadia-mediated destruction of TFs and TF regulators is however not known in many cases. This is exemplified by the inhibitors of the TGF-β and BMP pathways Smad 6 [128] and Smad7 [129]. Interestingly however, Arkadia can recognize the polySUMOylated transcription factor Nrf2, a master regulator of the anti-oxidant response system, and enables its stabilization after promoting its ubiquitylation by K48-linked ubiquitin chains. This observation is surprising, as K48-linked ubiquitin chains are principally known for their ability to address proteins to the proteasome [130]. Additionally, Arkadia [131], like RNF4 [119,120], can trigger SUMO-dependent ubiquitylation and degradation of the PML component of PML bodies, whose many roles are still ill-defined, but which are known to affect cell transcriptional regulation in case of dysregulation [132]. The mechanisms whereby the two StUbLs entail PML degradation are however different [131].

### 2.6. Can Transcription Involve Group SUMOylation?

An important lesson of recent functional- and large-scale genomic and proteomic studies is that SUMOylated proteins, at least for the most abundantly modified ones, are often, not only functionally interconnected within the same pathway, but also capable of physical association ([37,38] and references therein). This strongly supports the notion that they can undergo SUMOylation in a concerted manner within specific protein complexes/clusters in a process called “group SUMOylation”.

The concept of group SUMOylation has been first and particularly well exemplified in the responses to DNA damages, heat-shock and oxidative stresses [133,134,135,136]. It is based on the idea that SUMO/SIM interactions between the components of supramolecular complexes can serve as a glue to stabilize them. It also importantly asserts that not all SUMOylatable sites borne by their components need to be modified by SUMO at a given time for ensuring their stability [133,134,135,136]. Moreover, it is important to consider that SUMOylated components of these complexes can attract additional SIM-bearing proteins or -complexes. In particular, as certain components of the SUMOylation machinery harbor SIMs [10,37], SIM/SUMO interactions may also amplify SUMOylation of these complexes/clusters by retaining more firmly, or attracting more efficiently, SUMOylation enzymes at their levels. An implicit consequence of the group SUMOylation concept is that, if formation of complexes is dependent on the SUMOylation of certain of their components, then deSUMOylation should prove a rapid way to disassemble them. This points to a major role for, not only the SUMOylation machinery, but also for SUMO isopeptidases in the concerned biological processes and signaling pathways.

Although often evoked, the role of group SUMOylation has hardly been investigated formally and experimentally in transcription so far. There are however two good reasons to think that it actually applies to transcription. First, it is well established that many transcriptional actors work in complexes and that many of them are SUMOylated in vivo [2,3,4,37,38]. Second, Cossec et al. [137] have recently accumulated evidence that SUMO safeguards cell identity of mouse somatic and pluripotent/multipotent cells via acting as a glue stabilizing key transcriptional determinants of somatic and pluripotent states at gene enhancers. In a subsequent study [87], they also reported that the repertoires of SUMOylated substrates dramatically differ between somatic mouse embryo fibroblasts (MEFs) and embryo stem cells (ESCs), pointing to strikingly different roles of SUMOylation in transcriptional control in these two cellular contexts. Thus, in MEFs, SUMO-2/3 target proteins were largely found to be transcriptional regulators driving somatic enhancer selection, i.e., were largely associated with transcriptional activity. In contrast, in ESCs, SUMO-2/3 modification was largely observed on highly interconnected repressive chromatin complexes, which was proposed to prevent chromatin opening and transitioning to totipotent-like states. Future comparable genome/modifome/proteome-wide studies and/or more targeted functional studies in other biological setups are however still necessary to establish formally that group SUMOylation is a wide-spread mechanism essential for transcriptional control.

### 2.7. Can SUMO-Dependent Phase Separation Contribute to Transcriptional Regulation?

An important issue obviously relates to the molecular reasons and consequences of group SUMOylation. One of them is liquid/liquid phase separation, a mechanism currently the object of intense research which underlies intracellular organization through the formation of membrane-less organelles and -substructures [138]. In this general context, one attracting possibility would be that group SUMOylation facilitates transcription via promoting the formation of intranuclear condensates increasing locally the concentration of pertinent transcriptional actors. Along this line, it is of note that (i) Pol II most likely contributes to gene expression regulation through phase separation and (ii) multiple transcription factors and/or coactivators can form phase-separated supramolecular assemblies on enhancers with potential impact on gene activity [139,140,141]. Along this line, SUMO has recently been implicated in phase separation phenomena that would involve interactions between multiple SUMOylated- and SIM-bearing proteins (certain of them possibly presenting both characteristics) in several occasions. Thus, SUMO/SIM interactions were proposed (i) to participate in liquid-like droplet formation associated with PML body assembly [52,142], (ii) to be essential for phase separation of the cytoplasmic polyadenylation element-binding protein 3 (CPEB3), which is instrumental for translation inhibition of its target mRNAs via localizing them into the cytoplasmic P bodies [143] and (iii) to be crucial for the clustering of telomeres in phase-separated nuclear condensates in certain cancer cells [144,145]. Future large-scale or genome-wide research will consequently have to address whether similar phase-separation mechanisms apply to SUMO for promoting the formation of functional chromatin subcompartments involving important gene transcription regulatory elements (i.e., promoters, enhancers and domains controlling chromatin 3D organization) together with essential actors of gene transcription.

## 3. Regulation of Transcription by SUMO: Not Just Gene Repression

Thus far, the relationship between SUMOylation and transcription has principally been addressed in the case of protein-coding genes, though not exclusively (see below). Moreover, SUMOylation has most often been associated with transcriptional repression in the literature [1,2,3,4,5] despite rather early reports of SUMO-dependent gene activation. To date, the documented cases of transcriptional limitation, -down-regulation or even shut-off are, by far, the most numerous. However, that SUMOylation can positively impact transcription must be neither disregarded nor underestimated.

The long-standing and dominant assumption of transcriptional repression by SUMO was, in fact, largely the consequence of a historical bias, as most initial studies (strengthened by many later reports) on transcription regulation by SUMO pointed to repression of gene expression [2,3,91,146]. For example, pioneering reports indicated that inhibiting the SUMO pathway via depletion of either the SUMO-conjugating enzyme Ubc9 or SUMO itself or, alternatively, overexpressing SUMO isopeptidases, enhanced the transcription of large groups of genes [147]. Furthermore, a transcription-repressive action of SUMO was also initially observed in transfection assays using chimeric proteins involving fusions of SUMO-1 or -2 to the DNA-binding domain of the Gal4 transcription factor [148] and SUMOylation of most of the >200 TF studied for this PTM till 2017 was associated with reduced or repressed transcription [3,4].

The picture of gene regulation by SUMO, however, progressively appeared more contrasted when data supporting a positive role for SUMOylation in transcription activity/activation begun to accumulate. Thus, and among other examples, SUMOylation was shown to facilitate the recruitment of the basal transcription machinery, to prevent that of transcriptional co-repressors in certain situations and at certain genes [71,116,149,150,151,152], as well as to facilitate the recruitment of transcriptional coactivators such as the CBP/p300 lysine acetyl transferases [113,151,153] (see below for more details). Moreover, SUMOylation of a number of transcription factors was shown, -or at least proposed-, to potentiate their transcriptional activity, in certain settings and sometimes at a restricted number of target sites. Among others, these included Pax6 [154], p45/NFE2 [155], GATA-1 [156], the AR nuclear receptor [157], GATA-4 [158], Smad4/DCP4 [159], NFAT1 [160], PEA3 [161], HSF1 [162], HSF2 [163,164], JunB [165], Oct4 [94], Bcl11b [166], p53 [167], TCF4 [113], Ikaros [150] and FoxM1 [55]. It can, however, not be excluded that these effects might be context-, timing- and/or SUMO paralog-dependent. For instance, SUMOylation of a TF such as p53 was demonstrated to both up- and down-regulate its transcriptional activity, depending on the experimental setting [168,169] and there is no reason not to believe that the same may apply to other TFs. Another example is that of HSF1, which is the major stress-responsive transcription factor (out of a family of 3 members) that allows cells to cope with heat shock (HS) and survive if it is not too long and/or too strong. HSF1 has been shown to be modified by both SUMO-1 and SUMO-2/3, albeit possibly not with the same timings after HS onset with the modification by SUMO-1 occurring during HS and that by SUMO-2/3 during the recovery period following a short HS [162,170,171]. Modification by SUMO-1 was initially proposed to stimulate its transcriptional activity to induce heat shock protein genes, which code for chaperones essential for resoling/avoiding toxic protein aggregates caused by HS [162]. This conclusion was however questioned by others [112,171]. In contrast, modification by SUMO-2/3 is responsible for down-regulating HSF1 transcriptional activity to allow stressed cells to return to their initial non-stressed state [170].

Other reports have also consolidated the idea that gene regulation by SUMO is not necessarily a matter of mere on/off transcriptional switch. Along this line, a first series of observations showed that SUMOylation of TFs can occur concomitantly to the activation of their target genes to buffer the transcriptional output and, thereby, most probably avoid unphysiological and detrimental overexpression of the activated genes [149,172,173,174]. This supported the important concept that SUMOylation may be involved in *limitation* rather than just *repression* of transcription to ensure optimal gene expression, at least under certain conditions and for certain genes. Genome-wide experiments revealing a positive correlation between highly active gene transcription and SUMOylation of promoter-bound proteins also showed that this transcription limitation effect does not only apply to genes induced in response to stimulatory cues but also to numerous cellular genes in cells grown under standard conditions to maintain homeostasis [175].

Another important concept, applying at least to an acute stress that can entail cell death such as HS, is that SUMOylation of factors binding to gene promoters may be essential to avoid transcriptional down-regulation and, thereby, *maintain maximal expression* of certain genes essential for cell survival [22]. This effect is however not seen at all genomic loci [22,176,177] (also see below for more details). It must also be considered that, within the same signaling pathway leading to cell transcriptome reprogramming, factors with positive actions and regulators with negative effects can both be SUMO substrates [10]. As a consequence, the final transcriptional output in response to environmental cues results from a delicate equilibrium between different SUMOylation events occurring on different substrates exerting antagonistic effects within these pathways and/or the precise timing of these events. 

Making the picture more complex, it must also be considered that the same transcriptional actor can be SUMOylated/deSUMOylated at several stages of its life. The reasons for this can be different depending on the biological process concerned or, on the contrary, can potentially cooperate towards a common aim. Additionally, SUMOylation pathway components can collaborate with TFs independently of any SUMOylation activity. Elk-1, one of the first TFs whose SUMOylation has been studied, illustrates the latter two points as (i) its SUMOylation can down-regulate its transcriptional activity [147] and control its nucleo-cytoplasmic shuttling [178] and (ii) the SUMO E3 PIASx can act as a coactivator of Elk-1 by facilitating its derepression following a SUMOylation-independent mechanism [179].

That SUMOylation can be associated with transcriptional activity/modulation/limitation should however not hide that SUMO is also crucial for heterochromatin formation, which most often corresponds to a full transcriptional shut-off mechanism due to dense packing of DNA rendering it less, or no longer, accessible to both TFs and the transcriptional machinery. This point has already been mentioned above when addressing the issue of SUMO-2/3 modiform in mouse ESCs [87,137] and is developed below in a specific section.

## 4. Transcriptional Regulation by SUMO: Pol II-, but Also Pol I- and Pol III-Dependent Transcriptions

Pol II transcribes protein-coding genes and endogenous retroviruses together with a number of non-protein-encoding genes such as those for snoRNAs, snRNAs, miRNAs and lncRNAs whereas RNA Polymerase I (Pol I) is responsible for transcription of the multiple genes (rDNA) coding for the 45S precursor of the ribosomal 18S and 28S RNAs [91,92]. On its side, RNA Polymerase III (Pol III) transcribes tRNA genes (tDNA), 5S ribosomal RNA genes and several other small RNA genes such as those for the U6 snRNA involved in splicing or the 7S RNA of the signal peptide recognition particle [91,92]. Although most of the literature on gene expression regulation by SUMO deals with Pol II-dependent transcription, it must not be forgotten that transcription by Pol I and Pol III is also regulated by SUMOylation. A first hint supporting this idea is that high levels of chromatin protein SUMOylation were observed in ChIP-seq experiments at rDNA and tDNA in mammalian cells [22,175,177], as well as at tDNA and other Pol III -transcribed genes in budding yeast [149].

Concerning Pol I, it has been reported that SUMOylation limits Pol I-dependent transcription of rDNA in human cells cultured under standard conditions [175,180]. Recently, an indirect mechanism dealing with the coordination of Pol II- and Pol I-mediated transcription has been unveiled, as the authors show that the primary effect of SUMOylation is to reduce the expression of transcription factors (such as UBF and c-Myc, which are produced in a Pol II-dependent manner) binding to rDNA promoter regions [180]. As already evoked, global mass spectrometry analysis of protein SUMOylation has long revealed that, not only Pol II, but also Pol I- and Pol III basal machinery components are SUMOylated in vivo in *Saccharomyces cerevisiae* [181]. In addition, the RPA34 subunit of Pol I undergoes deSUMOylation in human cells in response to heat shock or treatment with the DNA-alkylating agent methyl methane sulfonate (MMS), even though the molecular consequences on Pol I activity were not addressed [59,62]. In contrast, a direct effect of SUMOylation on Pol I-mediated transcription was clearly documented in *Trypanosoma brucei* where Pol I, not only transcribes rDNA, but also the active gene of the variant surface glycoprotein (VSG), which allows the parasite to escape the immune response of the infected host [182]. This SUMOylation is associated with a positive effect on VSG gene transcription in bloodstream trypanosomes. Thus, SUMOylation of chromatin at the active VSG locus was shown to be required for efficient recruitment of Pol I in a SUMO E3 ligase (TbSIZ1/PIAS)-dependent manner, suggesting that protein SUMOylation facilitates the accessibility of additional transcription factors [183]. Moreover, SNF2PH, a SUMOylatable plant homeodomain (PH)-bearing transcription factor, was reported to be upregulated in the bloodstream form of the parasite and enriched at active VSG genes. More precisely, SUMOylation of SNF2PH promotes its recruitment to the VSG promoter, where it is instrumental to maintain Pol I and, thereby, efficient transcription of the VSG gene [184].

As far as Pol III-mediated transcription is concerned, SUMOylation of tRNA genes has been associated with limitation of transcription in mammalian cells [175]. Moreover, recent evidence has accumulated in the yeast *Saccharomyces cerevisiae* showing that SUMO may regulate it at multiple steps and in different ways. Thus, SUMO promotes the activity of Pol III under standard growth conditions, as depletion of the SUMO E2 reduces tRNA synthesis [185]. However, SUMO effects can be the opposite in response to various signaling events or stresses, including rapamycin treatment (which inhibits the TORC1 kinase complex of the mTOR pathway involved in metabolism control), nitrogen starvation, DNA alkylation by MMS and heat-, hyperosmotic- and oxidative stresses [3,186]. Mass spectrometry approaches indicated that each stress triggers a specific SUMO stress response (SSR) centered on proteins involved in translation and chromatin regulation during which certain of them become highly SUMOylated whereas others get deSUMOylated. Strikingly, whereas the various stress-specific SSR signatures were largely non-overlapping, all types of stresses were characterized by deSUMOylation of Pol III, which correlated with a decrease in tRNA synthesis. More precisely, in normally growing cells, the active conformation of the Pol III holoenzyme is supported by SUMOylation of its subunits Rpc82, Ret1, Rpc37 and Rpc53 whereas, in stressed cells, deSUMOylation of these proteins possibly leads to destabilization of the Pol III complex, though the mechanisms may differ, depending on the nature of the Pol III-transcribed genes. Along the same line, the human orthologs of Rpc82, Rpc37 and Rpc53 (RPC3, RPC5 and RPC4, respectively) were found SUMOylated in Hela cells with RPC3 and RPC4 being deSUMOylated upon heat shock and RPC4 and RPC5 upon treatment with the DNA-damaging agent MMS [59,62]. Functional studies are however still required to establish the biological significance of these observations.

Besides these studies, Wang et al. [187] have recently conducted a genetic approach to uncover functional relationships between Pol III and SUMO in budding yeast. They found that SUMO can repress Pol III, especially under conditions of greatly reduced activity due to either decreased expression or disabling mutations, that are conditions different from those used by Chymkowitch et al. [185] and Nguea et al. [186] presented in the previous paragraph. The point is of importance, as mutations in Pol III components (which were tested in the Wang et al.’s genetic assay in yeast) were recently found to cause neurodegenerative diseases in human. More precisely, Wang et al. observed that SUMO, ubiquitin and the Cdc48 AAA-ATPase segregase act in a linear way to repress Pol III-mediated transcription [187]. This pathway first implies SUMOylation of the Rpc3 subunit of Pol III in a Siz1 SUMO E3-dependent manner. Then, SUMOylated Rpc53 would trigger the ubiquitylation of the Rpc160 catalytic subunit of Pol III (and possibly other proteins) by the chromatin-associated Slx5-Slx8 SUMO-targeted ubiquitin E3 ligase. Finally, ubiquitylated Pol III complexes are recognized and disassembled by the cdc48 segregase, entailing proteasomal degradation of the Rpc160 subunit, thus clearing obstructed tRNA genes to allow transcription to resume.

SUMO can also repress Pol III transcription via other means, in particular via modification of the Maf1 regulator. The latter factor is a master repressor of Pol III-dependent transcription in response to a broad range of intra- and extracellular cues largely controlled by the mTOR pathway [188]. Though in simple organisms, such as yeast, its actions appear restricted to Pol III, it also regulates Pol I- and Pol II-mediated transcription in higher eukaryotes. It is a focus of attention, as it plays a potential targetable role in tumorigenesis control [188]. Rohira et al. have reported that SUMOylation of Maf1 by both SUMO-1 and -2 on a single lysine (K35) is instrumental to downregulate Pol III-mediated transcription of tRNA genes in mammalian cells [189]. Of note, this SUMOylation is independent of mTOR pathway-dependent phosphorylation of Maf1, which is the best-documented control of the activity of this protein. Interestingly, Maf1 SUMOylation level is regulated by the SENP1 isopeptidase. Moreover, a non-SUMOylatable Maf1 protein is defective in its ability to associate with Pol III and, thereby, its ability to facilitate the dissociation of Pol III from tRNA genes.

Thus, taken together, the above data highlight a role for SUMO in the control of Pol I- and Pol III-mediated transcriptions. Nevertheless, much remains to be studied to fully understand the most probably multiple roles of SUMO in these processes, including in cells subjected to different cues. It is also important to take into account that there exist cross-talks between the three eukaryotic RNA polymerase systems. In particular, they share a number of components and regulators and their respective actions can be coordinated by cell signaling [91,92]. It will therefore be paramount to study how SUMOylation can interfere with these cross-talks and coordinations. 

## 5. The General Lessons of Genome-Wide Studies of Chromatin SUMOylation

### 5.1. Where, When and What?

It is now firmly established that PTMs of chromatin- and chromatin-associating proteins at gene regulatory elements such as transcriptional promoters and enhancers are central for gene expression control [102,103,104,105]. They regulate a variety of mechanisms that include alterations in the biochemical properties of chromatin histones and non-histone structural proteins, regulation of transcription factors and -cofactors activity, control of RNA polymerases machineries, as well as that of the regulatory complexes cooperating with them or factors crucial for chromatin 3D organization (see below). 

Due to the well-established role of SUMO in transcriptional regulation, various laboratories have addressed its distribution, as well as that of SUMOylation enzymes, on chromatin in the recent years. This was achieved using either large-scale standard chromatin immunoprecipitation (ChIP)- or genome-wide ChIP-seq approaches coupled to transcriptomic- and, sometimes, proteomic experiments. These were conducted in budding yeast [149,152,173] and mammalian (mouse and man) cells [22,175,187,190,191,192,193,194,195,196,197]. Most of them included a comparison of SUMO and SUMOylation enzymes distribution with those of histone marks specifying transcriptional activity or -inactivity and Pol II. Some also included comparisons with (i) the distribution of regulators of transcription or of chromatin architecture, (ii) ongoing transcription assayed genome-wide using the Gro-seq assay and/or (iii) open chromatin sites identified using DNAse I hypersensitivity assays (DNAse-seq), formaldehyde-assisted isolation of regulatory elements (FAIRE-seq) or accessibility to the Tn5 transposon (ATAC-seq). 

A first overall conclusion of these studies is that SUMO is widely distributed over the genome with increased concentration at a quite high number of discrete sites. These are found in the range of one to several tens of thousands, depending on the sequencing depth achieved (see Figure 2A for an example). Most ChIP-seq SUMOylation signals are broader than those of transcription factors that are usually sharp (1–200 bp range), but shorter than those of many histone modifications. They usually cover domains of 500–1000 bp in average, which is consistent with the idea that not just TFs are SUMOylated at DNA regulatory elements displaying SUMO signals. They can, however, sometimes be much longer or shorter. The latter case is best illustrated by tRNA genes that are particularly short genes (100 bp range).

Though SUMO signals can be found in intergenic regions as well as in exons and introns, many of these domains correspond to open regions of chromatin, which are assumed to recruit transcription- and/or regulatory factors at regulatory elements such as promoters, enhancers, silencers or chromatin domain insulators. The point is particularly striking in the case of Pol II-transcribed genes where a notable fraction (5–10%) of SUMO peaks lies just upstream the transcription start site (TSS) in the so-called nucleosome-free region [22,137,149,175,177] that recruits factors essential for transcription initiation, including certain TFs and the transcription preinitiation complex (PIC) [91,92] (see Figure 2B for an example). Of note, the three ubiquitous SUMO paralogs were found enriched at gene promoters (17% for SUMO-1 and 15% for SUMO-2/3) in human fibroblasts [175]. In the same cells, 67% of SUMO signals found at transcriptional start sites were associated with gene expression, in particular at histones-, protein biogenesis-, rRNA- and tRNA genes and correlated with high levels of RNA polymerase II (Pol II) recruitment [175]. However, at the same time, SUMO-1 and SUMO-2/3 signals were also observed at inactive, or poorly active, promoters marked by H3K27me3 [175], a histone modification usually associated with transcriptionally inactive chromatin. De novo sequence motif analysis at the latter SUMO peaks indicated an enrichment for the CTCF TF binding sites (BS), which was coherent with a previous observation reporting transcriptional repression associated with SUMOylation of CTCF [198].

With regard to transcriptional regulation, SUMO enrichment is, however, not limited to gene promoters. For example, putatively active enhancers were found highly SUMOylated in mouse embryonic fibroblasts [137]. SUMO-2 gets enriched at nucleosome-depleted DNA regulatory elements displaying classical histone marks of active chromatin in human osteosarcoma U2OS cells subjected to heat shock [22]. SUMO-2/3 also gets enriched at enhancer elements in human leukemia K562- and prostate cancer VCaP cells subjected to HS [177]. Interestingly, in U2OS cells, integration of ChIP-seq, transcriptomic and proteomic data suggested that SUMO-2 did not act as a direct activator or inactivator of the genes affected by the heat shock but rather as an acute stress response necessary for the stability of protein complexes involved in gene expression and post-transcriptional modification of mRNAs [22]. In contrast, in U2OS and VCaP cells SUMOylation was gained at many active promoters and enhancers where it appeared to restrict gene expression, possibly via promotion of Pol II proximal pausing downstream of TSSs [177]. This notion is discussed in details by others [176]). Whether the differences in gene transcription outcomes between the two studies were due to differences in cell types, nature of the response to HS and/or experimental conditions requires further investigations. 

Another important conclusion of these experiments is that the chromatin SUMOylation landscape can be partly or dramatically cell-specific [22,137,177] (Figure 2A) and is highly dynamic in response to extracellular cues or depletion of components of the SUMO pathway [22,175,177,190,191,192,193,194,195,196,197]. Importantly, such differences are associated with notable changes in transcriptional programs. As already mentioned, all of these genomic and transcriptomic investigations were, with no exception, important to strengthen the notion that SUMOylation can be involved in gene transcriptional activity, or gene transcriptional activation, and not just in transcriptional repression or limitation. Nonetheless, not all SUMOylation signals found on chromatin are associated with transcription regulation, as SUMOylation can also serve other aspects of chromosome biology that non-exclusively include genome stability control [199], overall chromatin architecture [5], heterochromatin dynamics [200], telomere maintenance [201] or guidance through mitosis and meiosis [202,203]. Moreover, SUMO ChIP signals reflect steady-state levels of SUMOylated proteins but give no clue on the dynamics of the SUMOylation process at a given time. This can be a limitation to certain genome-wide investigations when taking into consideration that binding dynamics of TFs can be linked more strongly to function than to TFBS occupancy [204]. 

A last important general conclusion of these studies is that the SUMO signals characterized in ChIP and ChIP-seq experiments are likely to be largely accounted for by SUMOylation events occurring within chromatin as (i) they are most often associated with SUMO E2 and E3 enzyme signals in the same ChIP/ChIP-seq experiments and (ii) RNAi-mediated elimination of the latter enzymes entails substantial reduction of SUMO signals in ChIP/ChIP-seq experiments [22,149,173,174,175,176,177,190,193,194,195,196,197]. Neither how SUMOylation enzymes are attracted to specific genomic loci, nor how this process is regulated are, however, known yet. These important questions deserve future investigations, even though the SUMO E3 Pias1 has already been suggested to be capable of binding to DNA on its own via its amino terminal scaffold [205]. It is also of note that SENP proteins have also been found associated to chromatin, suggesting that SUMOylation/deSUMOylation events can occur within this nuclear compartment. This, of course, does not exclude the possibility that proteins SUMOylated in the nucleoplasm can be recruited to chromatin and that SUMOylated proteins can be released from chromatin with their SUMO moiety still covalently attached to them.

### 5.2. A Common and Conserved Role for TF SUMOylation in TFBS Selection?

Concerning Pol II-dependent transcription, Rosonina [90] recently and interestingly raised the idea that a common, conserved and major (but non-exclusive) function of TF SUMOylation may reside in DNA binding-site selection control, irrespectively of whether TFs have positive or negative roles on transcription of their target genes. To reach this conclusion, he examined in detail four ChIP-seq-based studies conducted in different species. These addressed differential genome-wide binding to DNA of wild-type and of SUMOylation-deficient forms of the transcription factors MITF [206], glucocorticoid receptor (GR; [193]) and androgen receptor (AR; [207]) in human cells and Sko1 in *Saccharomyces cerevisiae* [195]. Strikingly, all SUMOylation-deficient TF variants were found bound to numerous additional non-specific sites (up to more than 2-fold), as compared to their wild-type counterparts. This, and the fact that SUMOylation may alter the affinity of at least certain TFs for their TFBSs [3], led the author to propose a model according to which (i) prior to their SUMOylation, TFs would initially bind to chromatin with reduced specificity, (ii) this (at least in part) spurious binding would ensure that all functional sites become bound and (iii), once the TFs are bound to DNA, SUMOylation would then increase binding specificity and promote the release of the TF from non-specific sites (Figure 3). This model obviously relies on the hypothesis that SUMOylation of TFs can occur when the latter are bound to DNA. As previously mentioned, there is already substantial evidence supporting this possibility. It must nevertheless be noted that, whereas SUMOylation generally destabilizes the association of TFs with non-specific sites according to the proposed model, in some cases, the modification must also stabilize transient TF–DNA interactions. This would explain the existence of a smaller subset of TFBSs, the recognition of which depends on intact SUMO motifs on the TFs.

Additional studies involving other TFs are still required to confirm on a broader experimental basis the notion of restricting TF binding to appropriate sites by SUMOylation. Moreover, how SUMOylation mechanistically controls TFBS selection constitutes an important ensuing question. In particular, it will be interesting to address whether SUMOylation influences TFBS selection by modulating protein-protein interactions with other nearby-binding TFs and/or with co-factors not binding to DNA and/or via promoting conformational changes within the TFs that alter their DNA-binding specificity and/or affinity. Finally, SUMO-dependent TFBS selection does not exclude other functions for TF SUMOylation (see below).

## 6. Molecular Mechanisms Whereby SUMOylation Can Regulate Pol II-Dependent Gene Expression

### 6.1. Technical and Biological Limitations when Studying the Regulation of Transcription by SUMO

As already illustrated above, the regulation of Pol II-dependent transcription by SUMO can follow a wide variety of mechanisms. However, caution is required when interpreting many past and current functional studies for at least two reasons. First, the experimental methods available to investigate the role of SUMO in gene expression control, although informative, present serious biases. Second, the SUMO field still suffers from the lack of essential tools that would secure entirely unambiguous conclusions. 

Thus, in in cellulo studies, the functional consequences of SUMOylation were most often addressed via altering the SUMOylation level of the proteins of interest using one, or several, of the following methods: (i) inhibition of protein SUMOylation by mutating either SUMO acceptor lysines (into arginines) or nearby residues, which may alter modification(s) by other PTMs essential for the function of the studied proteins, (ii) N- or C-terminal fusions of SUMO to these proteins, which neither recapitulate natural lysine SUMOylation and the dynamics of SUMOylation/deSUMOylation cycles they can be subjected to, nor reflect that only a minor fraction of the concerned protein is usually SUMOylated at a given time and (iii) altering the overall cellular levels of SUMOylation by either overexpressing SUMO or components of the SUMO pathway (E2, E3 and/or SUMO isopeptidases) or reducing their expression by RNAi or genetic knock-out of their genes, which also unavoidably affects SUMOylation of other cell components possibly interfering with the process under study.

Moreover, there is no easily and efficiently implementable technology currently available to generate antibodies specifically directed to the SUMOylated sites of SUMO substrates similar to those used to generate antibodies recognizing phospho-sites in phosphorylated proteins. The only such antibodies we are aware of are those raised against the SUMOylated c-Fos TF [174]. They have been generated using a branched synthetic oligopeptide where the C-terminal 6 amino acids of SUMO are chemically bound to a peptide containing the SUMOylated lysine of c-Fos. Such antibodies would, for instance, be particularly precious to analyze precisely SUMOylated transcriptional actors in ChIP and ChIP-seq experiments or for conducting diverse biochemical/proteomic/modifomic investigations. In addition, the fine transcriptional mechanisms regulated by SUMO were usually investigated on a limited number of genes and rarely at the whole genome scale. Moreover, most of the SUMO-dependent transcriptional mechanisms identified so far were studied at the level of gene promoters and little at other gene regulatory elements such as enhancers, silencers or sites controlling chromatin 3D organization/dynamics. Therefore, system-wide functional studies are still required to assess whether the scope of the mechanisms identified thus far is limited or general, as well as when and where they apply in living organisms.

Several categories of transcriptional mechanisms regulated by SUMO are (non-exhaustively) considered hereafter. They illustrate that there is no univocal mechanism whereby SUMO controls transcription (also see [1,2,3,4,5,176,200]). Moreover, the mechanistical consequences of certain SUMOylation events can be multiple.

### 6.2. SUMOylation-Dependent Alteration of the Intrinsic Properties of TFs

TFs are central in the nucleation and firing of the transcriptional act or, on the contrary, transcription repression. As illustrated several times above, there is ample evidence that SUMOylation of TFs can alter their intrinsic properties with very different transcriptional outcomes, depending on the TF and on the experimental setting. These properties non-exclusively include subcellular localization, metabolic stability, interference or cooperation with other PTMs, homo- or heterodimerization, binding to- or clearance from DNA and/or association with chromatin and ability to interact with other protein partners (see next section) (Figure 3).

As described above, an important lesson of genomic/transcriptomic studies is that SUMOylation of TFs (and other proteins) can largely occur within chromatin [22,175,177,190,191,192,193,194,195,196,197]. This is obviously not exclusive of possible TF SUMOylation at other locations in the cell, as there exist multiple examples of control of TF subcellular localization by SUMO. These include the regulation of cytoplasmic retention, nucleo-cytoplasmic shuttling or association with nuclear substructures such as PML bodies, the final effect of SUMOylation being enhanced or decreased access to DNA [8,10,147,208,209,210,211].

If SUMOylation within chromatin may largely be associated with TFBS selection by TFs (see above and [90]), there is also ample evidence that it can also alter DNA-binding efficacy or facilitate the release of TFs from their binding sites once their transcriptional part played [172,173,174]. Concerning binding to DNA, there are, for example, indications that SUMOylation of the TF Sp1 by SUMO-2/3, but not by SUMO-1, decreases its ability to bind to the βB1 cristallin gene promoter in developing ocular lens cells and, thereby limits its ability to stimulate transcription [212]. There is also in vitro and in vivo evidence that the same applies to the TFs Prox1 in its regulation of the VEGFR3 gene [213], GATA-1 in hematopoiesis [156] or FOXA1, which is a pioneer TF cooperating with steroid receptors [207]. Alternatively, SUMOylation can alter TF homo- or heterodimerization with impact on transcriptional activity. The TF FoxM1, which is a key regulator of cell cycle progression, illustrates this possibility. Its SUMOylation peaks during G2/M, which is essential for proper progression through mitosis. This SUMOylation blocks FoxM1 dimerization and, thereby, relieves autorepression and activates transactivation activity to stimulate transcription at target genes [55]. Others have found that SUMOylation may repress transactivation by FoxM1 [208,214]. But this may be to technological biases linked to the use of FoxM1 fusion proteins with Ubc9 or SUMO and ectopic overexpression of Ubc9 [55]. 

On the opposite, SUMOylation can also enhance binding to DNA. This is the case for Oct4, a TF affecting the fate of certain progenitor- or cancer cells [94]. This example is interesting in that the effect of SUMOylation are multiple, as SUMOylated Oct4 also shows increased metabolic stability and stronger transactivation activity [94]. That SUMOylation may have multiple effects on TFs is, in fact, not unusual. For example, SUMOylation of ATF7 delays its entry into the nucleus, blocks binding to its TFBSs at target gene promoters and inhibits interaction with the general transcription factor TAF12 [215].

Besides this, SUMO-dependent clearance from chromatin can be achieved by simple removal of TFs from the DNA template or protein destruction. This helps adjust transcription to the right level, control the duration of transcriptional responses, reset promoters and/or recycle TFs for other transcriptional actions when the latter are not proteolysed [173,174]. This, however, does not exclude other roles for SUMO before SUMOylated TFs are released from chromatin (see below). An illustrative example is that of the TF Gcn4 in *Saccharomyces cerevisiae* where Gcn4 undergoes SUMOylation after binding to its target promoters and after Pol II has been recruited owing to this TF binding event [173]. This facilitates Gcn4 removal from chromatin during ongoing transcription and, thereby, prevents excessive accumulation of Pol II at the concerned promoters. For clearance, SUMO activates the phosphorylation of Gcn4 by the Srb10 kinase of the mediator complex, which triggers Gcn4 proteolysis through the ubiquitin/proteasome pathway [172,173]. Moreover, coordination between SUMOylation of Gcn4 and that of the corepressor Tup1 (and probably other factors) are necessary to dampen transcription [216]. Along the same line, SUMOylation of the TF Fos facilitates its clearance from chromatin at the level of, at least, certain of its target genes in mammalian cells [174]. Additionally, the fact that SUMO isopeptidases were found at many transcriptional regulatory elements in the genome using ChIP-seq, together with a number of functional studies, also raises the possibility of deSUMOylation of TFs as an important regulator of gene expression. Finally, SUMOylation of TFs themselves can be a complex phenomenon involving multiple additions of SUMO moieties, or the formation of polySUMO chains, whose outcomes may also be diverse, as discussed earlier in the case of the StUbLs RNF4 and Arkadia [119,120,121,126,127,130,131].

### 6.3. Regulation of Interactions between Transcription Factors and Transcription Co-Regulators by SUMO

Various studies have indicated that SUMO can facilitate, or, on the contrary, inhibit the recruitment of transcriptional co-regulators by TFs (Figure 3). The final outcomes of these interactions can be up- or down-regulation of gene expression, depending on the gene and on the context, as exemplified hereafter.

Acetylation of lysines lying in the N-terminal tail of histones is considered as a PTM associated with transcriptional activity/activation. It is deposited by histone acetyltransferases (HATs; also called lysine acetyltransferases or KATs), which also acetylate other proteins, some of them being implicated in transcription [217]. Histone deacetylase (HDACs) remove these acetyl groups and are usually regarded as negative regulators of transcription. Since HDAC2 was shown to be the first corepressor to be recruited by a SUMOylated transcription factor [147] (see above), the recruitment of HDACs by SUMOylated transcription factors has been documented several times and associated with transcriptional down-regulation [1,218,219,220,221,222,223,224,225,226,227,228,229].

The picture of the actions of SUMO on HDACs is however more complex than mere HDAC recruitment by SUMOylated TFs owing to SIM/SUMO interactions. Thus, when they are part of multisubunit transcriptional regulatory complexes, HDACs can be recruited indirectly via other SUMOylated proteins, as discussed later. Moreover, in certain cases, TF SUMOylation limits rather than enhances HDAC recruitment (Figure 4). This is the case of HDAC3 whose interaction with the basal transcription factor TFII-I binding gene initiator elements in promoters is impaired upon SUMOylation of the latter, entailing less efficient transcription of genes such as Fos in a liver cancer cell line [71]. This is also the case of Ikaros, the SUMOylation of which attenuates its transrepression activity via disruption of its interaction with both HDAC-containing- and HDAC-non-containing repressor complexes [150]. Of note, certain HDACs can themselves be SUMOylated, which may have different outcomes. One of them is alteration of enzymatic activity. An illustrative example is HDAC2, whose SUMOylation can stimulate NF-κB transcriptional activity, at least under certain conditions [230]. Conversely, SUMOylation of HDAC2 has been found to increase interaction with and deacetylation of the TF p53 during genotoxics stress, leading to reduce recruitment of p53 at its target genes and, consequently, attenuation of DNA damage-induced apoptosis [231]. Another consequence can be increased degradation, which is associated with higher gene transcription [232]. The effects can also be different, depending on the conditions. For example, HDAC1 plays a dual role in signaling by the myogenic TF MyoD: enhancement of MyoD deacetylation in its basally SUMOylated state in undifferentiated myoblast and dissociation of HDAC1 from MyoD during myogenesis [233]. Finally, and adding another layer of complexity to the cross-talk between SUMO and HDACs, the members of the HDAC class II family (HDAC-4, -5 and -7) have been proposed to display a SUMO E3 activity for certain substrates, at least under conditions of overexpression [25,26,27]. This might constitute a feed-forward process to enhance SUMO-mediated histone deacetylation and, consequently transcriptional repression.

SUMOylation can also regulate the function of certain lysine-specific methyl transferases (KMTs). They transfer methyl groups to lysines of a variety of proteins, including histones [234,235]. Histones can be methylated at various lysines of their N-terminal tail, which, like acetylation is instrumental for nucleosome compacting/decompacting and, thereby, transcription regulation. Depending on the histone and the lysine residue modified, methylation of lysines can be associated with transcriptional activation or repression/silencing [102,103,104,105]. Like HDACs, certain KMTs can also be recruited by TFs upon their SUMOylation through SIM/SUMO interactions. For example, SUMOylation of the Sharp-1 repressor is necessary for inhibiting the differentiation of myoblasts into skeletal muscle. One key mechanism at play is the recruitment of the KMT G9a by SUMOylated Sharp-1 for full transcriptional repression of muscle-specific genes [236]. However, as for HDAC2, the SUMO-dependent functions of G9a can be multiple. For instance, together with its anti-differentiation action in skeletal myoblasts, G9a exerts a pro-proliferation effect. The latter is, not only dependent on G9a own SUMOylation, but also paradoxically associated with a positive effect on transcription. Thus, SUMOylation of G9a permits interaction with the histone acetyl transferase PCAF. This promotes the association of PCAF with the pro-proliferation transcription factor E2F1 at its target genes where an increase in H3K9 acetylation is observed [237]. An important issue is to characterize the molecular mechanisms responsible for the differential gene-specific actions of G9a.

Other corepressors can be recruited by SUMOylated TFs. Some well-documented examples include NCoR1, CoREST1 and SMRT. These transcriptional coregulators interact with other nuclear proteins, including various HDACs and KMTs or lysine-specific demethylases (KDMs). They play a key part in modulating the activity of various members of the TF superfamily of nuclear receptors by modulating epigenetic marks on chromatin [238,239]. Thus, SUMOylation of the human PPARα TF can down-regulate its transcriptional activity via recruitment of NCoR1 and the histone deacetylase HDAC3 [227,228]. Of note, NCoR1 is itself SUMOylated, which enhances its transcription repression potential [240]. Along the same line, direct binding of CoREST1 to SUMO-2/3, but not SUMO-1, was shown to contribute to gene-specific repression by a complex also containing the KDM LSD1 and a HDAC activity via a SUMO/SIM interaction [219]. Moreover, the TF Gfi1, which is involved in hematopoiesis differentiation determination, undergoes SUMOylation implicating the SUMO E3 PIAS3 [220]. This is necessary to recruit a CoREST1- and LSD1-containing complex and turn off the transcription of genes such as Myc to enable granulopoiesis [220]. Transrepression by CoREST1 might be compromised by its own SUMOylation [241] (which is, however, not seen by other authors; see [242]), and several subunits of the LSD1/CoREST1/HDAC complex (including HDAC1, LSD1, ZnF198, BRAF35 and CtBP) have been reported to be SUMOylated. In particular, that of BRAF35 appeared necessary for full repression of neuron-specific genes in non-neuronal cells and for occupancy of the CoREST1/LSD1 complex at its gene targets [229]. The case of the glucocorticoid receptor (GR) is also interesting to consider. GR can both transactivate and transrepress a variety of genes. Transactivation occurs through direct binding to GR-responsive DNA elements (GREs), whereas transrepression can occur through two different mechanisms. One is direct binding to so-called inverted repeated negative response elements (IR nGREs) via DNA-protein interactions. The other is indirect via protein-protein interaction-mediated binding to other TFs such as NF-κB, AP-1 or STAT3 bound to their respective TFBSs. In both cases, transrepression relies on former SUMOylation of GR, which permits the recruitment of a SMRT/NCor1-HDAC3 repressing complex [221,222].

Though less frequent, recruitment of coactivators by SUMOylated TFs to promote transcription has also been reported (Figure 4). This is, for example the cases of (i) the TF Bcl11b whose SUMOylated form can recruit the HAT p300 to derepress genes it otherwise represses during T-cell development [113], (ii) the CLOCK1/BMAL1 transcriptional complex, which, upon SUMOylation of its BMAL1 moiety during the resetting of the circadian clock, recruits the HATs CBP/p300 [153], and (iii) the TF RORγT, which recruits the HAT KAT2A by during the differentiation of T_H_17 cells [243]. Alternatively, coactivators can be recruited in a SUMO-dependent manner by other coregulators to promote the transcription of certain genes. As already mentioned, this is the case of the HAT PCAF, which is recruited by the SUMOylated KMT G9a at E2F1-dependent genes in proliferating myoblasts [237]. In another scenario, p300 was proposed to serve as a transcriptional co-activator of the c-Myb proto-oncoprotein indirectly through bridging of the two molecules by the SUMO E3 PIAS1 [151]. Finally, if SUMOylation of Sp1 by SUMO-2/3 reduces binding to DNA as presented above, modification by SUMO-1 on the same lysine residue has an opposite transcriptional outcome, as it facilitates p300 recruitment and enhances target gene expression, indicating paralog specificity [212].

All the above discussed examples highlight the concept that SUMO, not only regulate protein-protein interactions via SIM-bearing protein recognition, but also helps organize multi-subunit complexes involved in transcription regulation. They also pose the question of coordination of multiple SUMOylation events within and between these supramolecular complexes, possibly via group SUMOylation.

### 6.4. Regulation of Transcription Initiation and Elongation by SUMO

Pol II-dependent transcription is usually subdivided in four stages: initiation, pausing, elongation and termination. The fact that high levels of SUMOylation are observed in gene promoter regions, especially just upstream of TSSs, suggests that the former three stages may be governed by SUMOylation [22,108,149,152,173,175,177,190,191,192,193,194,195,196,197] (Figure 5). Moreover, most SUMOylation studies that have been conducted on specific transcription factors and co-regulators reinforce this idea, as they have essentially been carried out in genes’ TSS vicinity (see above). However, the effects at gene promoters are not limited to local epigenetic changes or chromatin topology reconfiguration. As already mentioned, different components of Pol II itself, as well as general transcription factors (GTFs) or factors involved in transcriptional pausing have been characterized as SUMOylated in different proteomic analyses [37,38]. At this stage, it is also interesting to note that the function of a TF is not necessarily limited to DNA binding followed by control of co-regulator recruitment. It may also include enzymatic actions. Thus, the human TF hDREF is essential for transcription of a number of house-keeping genes, including those coding for ribosomal proteins where it binds to specific DNA motifs located just upstream the TSSs [28], i.e., at a place where high SUMOylation levels were found in ChIP-seq experiments [22,108,149,152,173,175,177,190,191,192,193,194,195,196,197]. There is evidence that hDREF possesses a SUMO E3 activity towards the Mi2α subunit of the transcription-repressive ATP-dependent nucleosome remodeling and deacetylation complex NuRD. Thereby, it induces the dissociation of NuRD from the gene loci, permitting higher expression [28]. To our knowledge, the implication of SUMO in transcriptional termination has not been investigated yet.

During the transcription initiation step, the GTF complex TFIID binds to the proximal region of promoters, where it triggers the loading of both other GTFs and Pol II to form the so-called preinitiation complex (PIC). Shortly after initiation, Pol II pauses in a transcription-competent state a few dozens of nucleotides downstream of TSSs in approximately 30% of genes. This pausing is promoted by the multi-subunit complexes NELF and DSIF, which halt RNA polymerization by Pol II. This pause can then be released by the P-TEFb factor made of two subunits, cyclin T and the kinase cdk9. This serves to fine tune gene expression in response to changing cellular environments. Upon appropriate stimuli, P-TEFb phosphorylates NELF, DSIF and the C-terminal domain (CTD) of Pol II on Ser2 of the heptad repeats it is made of. This releases NELF from the PIC, turning DSIF into a positive elongation factor and stimulating the elongation activity of Pol II. Interestingly, TFIID has long been reported as a SUMO-regulated factor via modification of several of its subunits. These include TAF5, which modulates its binding to DNA, TAF1, TAF8, TAF12 and the TATA-binding protein TBP [244]. Of note, other GTFs were found SUMOylated, though the consequences of their SUMOylation were not addressed [245]. The study of the TF NZFP also supports the implication of SUMO in transcription initiation control. TF NZFP is a transrepressor required for early development during gastrulation of *Xenopus laevis* embryo. Its transrepression activity depends on NZFP interaction with TBP (TATA-binding protein), a factor typically binding just upstream of TSSs, which is promoted by NZFP SUMOylation and leads to inhibition of basal transcription complex formation [246].

Besides this, SUMO might also participate in transcriptional pausing regulation. First, heat-shock increases the SUMOylation of subunits of DSIF (hSpt5), NELF (A, B, C/D, E) and P-TEF (cdk9), which is associated with changes in gene expression [176,177]. Second, it was shown that SUMO limits global cell transcription by controlling the level of cdk9 SUMOylation. This prevents cdk9 interaction with cyclin T and, thereby, the formation of an active P-TEFb complex essential for productive transcription elongation [247]. This mechanism is crucial for repressing HIV-1 provirus transcription and promoting viral latency in infected cells [248]. In this case, it involves the SUMO E3 TRIM28 (also known as TIF1β for transcriptional intermediary factor 1 β or KAP1 for KRAB-associated protein-1) [248].

### 6.5. SUMO-Dependent Configuration of Nuclear Substructures

The epigenetically-controlled local, long-distance or global 3D organization of chromatin can be dynamically affected in response to numerous cues [249,250]. The consequences are important for gene expression since reorganization of chromatin can make chromatin transcription-permissive or -non-permissive [249,250]. Transcription can even be stably closed, as occurs in constitutive heterochromatin (discussed in next section).

Much work has now shown that SUMO can affect chromatin dynamics at different levels, which is not surprising when considering the large number of chromatin proteins found SUMOylated in proteomic/modifomic studies [37,38]. A first level concerns structures easily identifiable under the microscope. Thus, it has long been shown that nuclear integrity itself is dependent on SUMO [251]. SUMO also affects nuclear substructures such as chromosomes [252], the nucleolus [252], PcG bodies [5,252,253], PML bodies [52,136] or heterochromatin [200]. 

Interestingly, PML- and PcG bodies are, not only nuclear substructures showing SUMO-dependent nucleation or maintenance and association with SUMOylation pathway components, but also SUMOylation centers impacting chromatin organization and transcriptional activity. For example, the histone H3.3-specific chaperone complex DAXX/ATRX traffics between PML bodies and DNA, with SUMOylation of DAXX being required for localization at PML bodies [254]. Moreover, a number of TFs and transcriptional regulators were also found associated to PML bodies in a SUMO-dependent manner [10,52]. PcG bodies, on their side, are foci of Polycomb group proteins, the latter forming two major complexes, PRC1 and PRC2, which are developmentally regulated and function as hubs for transcriptional repression [50]. Noteworthy, one of the components of PRC1, Pc2/CBX4, acts as a SUMO E3 [255,256,257,258] for a number of proteins. These include the DNA methyltransferase Dnmt3a, which is important for silencing of many genomic loci [259], certain TFs such as HIF1α [260], a transcriptional corepressor such as CtBP1 [255] or proteins with multiple impacts on gene expression and chromosome architecture such as CTCF [198]. Moreover, Pc2/CBX4 own SUMOylation and deSUMOylation was reported to be essential for lineage specification during mouse early development. It regulates PRC1-mediated selection, binding and release of PcG target genes [261]. Moreover, SUMOylated PRC1 controls histone H3.3 deposition and genome integrity of embryonic heterochromatin in a process involving the histone chaperone DAXX/ATRX [262]. 

At a much lower scale, TFs binding to gene promoters can SUMO-dependently recruit transcriptional coregulators that are chromatin remodelers or post-translational modifiers of chromatin-associated/-constituting proteins affecting nucleosome compaction/decompaction or movements (see above). This also argues for an important role of SUMO in controlling local chromatin configuration in relation with transcriptional competence/incompetence. That many of these coregulators were found themselves regulated by SUMOylation strengthens this idea (see above). In this respect, the case of the poly(ADP-ribose) polymerase PARP1 is worth being noted. Upon heat shock, the SUMO E3 PIASy interacts with PARP-1 and enhances its multi-polySUMOylation at the HSP70.1 gene promoter [263]. This appears mandatory for fully activating its transcriptional potential, most probably through polyADPribosylation-dependent nucleosome decompaction. Once nucleosome remodeling is achieved, PARP-1 is degraded by the proteasome after ubiquitylation by the StUbl RNF4, allowing transcriptional firing [263]. 

There is additional evidence that SUMO plays a part in the control of chromatin 3C organization at different levels. 

### 6.6. Histone SUMOylation and Transcription

Histones are major actors of chromatin organization and their PTMs largely decide the level of nucleosome packaging by favoring or disadvantaging internucleosomal interactions as already discussed. Both mammalian [264] and yeast [265] histones have long been shown to undergo SUMOylation with however different modalities. In mammals, SUMOylation occurs predominantly on histone H4 and H3. Yet, the other core histones (H2A and H2B) [37,38,54,264], the linker histone H1 and the variant histones H2AX and H2A.Z playing a part in DNA damage responses are also concerned, though to a lesser extent [37,38,86,266,267,268]. In contrast, in budding yeast, all core histones are SUMOylated, sometimes at multiple lysines [265]. As with most of other histone PTMs, SUMOylation occurs in the N-terminal flexible tail of histones. It can therefore compete and/or cooperate with PTMs associated to transcriptional activity/inactivity. Due to their SUMO moiety, SUMOylated histones are likely to recruit SIM-bearing proteins influencing transcriptional competence of chromatin. Supporting this possibility and using an in vitro system based on chemical SUMOylation of human histones, Dhall et al. have reported that nucleosomes containing SUMOylated H4 could prevent chromatin compaction by inhibiting long-range internucleosomal interactions [269] and stimulate the demethylase activity of the KDM LSD1 following a mechanism dependent on a SIM located in CoREST1 [270]. To which extent and on which occasion this observation applies in vivo still remains to be studied. It is however likely that histone SUMOylation may have different roles, depending on the genetic locus and cell signaling. Thus, in initial studies, histone SUMOylation was shown to increase interaction with both HDAC1 and heterochromatin 1 γ (HP1γ), which was suggestive of a predominant transcription-repressive role for this PTM [264] (see next section). However, SUMO was also detected on acetylated histone H4 and the HAT p300 was shown to increase H4 SUMOylation [264], suggesting possible association with transcriptional activity. Along this line, the Ulp2 SUMO protease promotes transcription elongation through regulation of histone H2B polySUMOylation at gene promoters and subsequent phosphorylation of Pol II CTD in budding yeast [271]. The study of the RSC complex, still in *Saccharomyces cerevisiae*, also supports the idea that histone SUMOylation may affect chromatin configuration. RSC is an abundant, essential chromatin-remodeling complex of the SWI/SNF family involved in transcription and several other aspects of chromosome biology. It was recently shown that its interaction with nucleosomes is modulated, not only by H3K14 acetylation, but also by SUMOylation of H2B, which regulates genetic information accessibility [272].

### 6.7. SUMOylation and Organization of Topologically-Associating Domains

Metazoan chromatin is organized into spatially segregated regions called topologically-associating domains (TADs) defined by frequent intra-TAD contacts and infrequent contacts with neighboring regions [249,250]. Moreover, chromatin looping is at the basis of higher order chromatin structures such as TADs, as well as other functional structures such as enhancer-promoter contacts often aggregating in hubs within TADs [249,250]. A variety of genomic elements contribute to this dynamic organization, which is essential for proper gene expression and regulation. Among others, they include insulators, which establish distinct domains so that gene expression can be independently regulated in adjacent domains. They also comprise matrix attachment regions (MARs), which function as intranuclear anchor points for chromatin loops. 

Evidence has accumulated for SUMO controlling all these aspects of chromatin organization with impact on transcription [5,200]. A first example is that of CTCF, which is a zinc finger DNA-binding protein binding to >10,000 sites in mammalian cell genomes [273]. It displays activation, repression and insulator activities and also participates in regulating chromatin architecture by maintaining chromatin open, mediating long-range chromosomal interactions and organizing chromatin into TAD structures [273]. It can be SUMOylated in one of its functional domains showing both transcription and chromatin decondensation abilities, though the two functions do not necessarily work together [198,274]. SUMOylation does not seem to alter CTCF binding to DNA but reduces transcription activation and chromatin opening [198,274]. SUMOylation at CTCF-binding sites has also been proposed to be involved in maintenance of chromatin anchor complexes [176]. Interestingly, the SUMO isopeptidase SENP1 and the chromatin remodeler CHD3 were shown to interact and to jointly affect chromatin accessibility and gene expression at multiple genomic loci, showing cooperation between deSUMOylation and chromatin reorganization [194]. DeSUMOylated proteins were not identified but the genomic analyses conducted in this study raised the possibility that CTCF might be a target for SENP1 [194]. Along the same line, at least two protein components (Mod(mdg4) and CP190) of the *Drosophila melanogaster* gypsy insulator are SUMOylated, which attenuates chromatin insulator activity [275]. SUMOylation is necessary for these insulator proteins to colocalize in nuclear speckles named insulator bodies [276], as well as for enhancer blocking [277]. SUMOylation of drosophila CTCF (dCTCF) plays the same role [277]. Also worth of mention, PIAS1-dependent SUMOylation of the MAR-associated SATB2 protein was shown to modulate the expression of the immunoglobulin µ gene in preB cells through enhancing association of the µ immunoglobulin locus with MARs [278]. Along the same line, the scaffold-associated factor B 1 (SAFB1) displaying both DNA- and RNA-binding abilities is SUMOylated at promoters of ribosomal genes, which stimulates both binding of Pol II to gene TSSs and pre-mRNA splicing [279]. Finally, SUMO is also important for the formation, maintenance and turnover of PML bodies, as well as various of their functions besides serving as SUMOylation hubs [52,210]. For example, functional interactions between PML bodies and the MAR-interacting protein SATB1, which is a SUMO substrate [280], were shown important for regulating chromatin-loop-architecture and transcription at the MHC class I locus and, thereby, coordinated expression of a subset of MHC-I genes [281]. Moreover, several chromatin remodelers traffic between PML bodies and DNA in a SUMO-dependent manner. This is the case of the already cited histone chaperone complex DAXX/ATRX, which interacts with SUMOylated PML bodies through one of its SIMs and whose function is regulated by SUMOylation [282]. PML proteins may also have functions outside of PML bodies. In particular, they contribute to large-scale (mega-bases) heterochromatin organization through regulation of histone H3.3 deposition by DAXX/ATRX [283]. The role of SUMO in this process has, however, not been investigated yet.

### 6.8. SUMO in the Establishment of Heterochromatin

Partitioning of euchromatin and heterochromatin is an important layer of chromatin organization. Whereas euchromatin comprises transcriptionally active regions of the genome, heterochromatin corresponds to condensed forms of chromatin with poor, or no, transcriptional activity [284]. Heterochromatin is categorized into two major types: constitutive and facultative. The former is found in all cell types. It principally corresponds to repeat-rich and gene-poor regions of the genome such as those found around centromeres and telomeres. But it also includes other elements. These can be retroposons, including endogenous retroviruses (ERVs) in mammalian cells, and also transposons in other species, which are scattered in euchromatic domains, as well as some gene-poor regions. In contrast, facultative heterochromatin is cell- and/or signaling-specific and is usually associated with gene transcription repression that is sometimes reversible. Heterochromatin formation is largely determined by histone modifications that are recognized by other proteins instrumental for DNA packaging and, thereby, transcriptional shut-off at the concerned loci [5,50,200]. Thus, trimethylation of histone 3 lysine 9 (H3K9me3) is enriched in constitutive heterochromatin domains and is deposited by the Suv39h and SETDB1 families of histone methyl transferases in humans. These HMTs, and their homologues in other species, can self-propagate heterochromatin since they recognize H3K9me3 and methylate adjacent nucleosomes. H3K9me3 is recognized by the heterochromatin proteins of the HP1 family, whose dimerization brings adjacent chromosomes into closer proximity. In contrast, facultative heterochromatin is usually marked by trimethylation of histone H3 lysine 27 (H3K27me3) (but not only) and the presence of the Polycomb repressive complexes PRC1 and PRC2, which are responsible for H3K27me3 deposition [5,50,200]. 

Over the year, evidence has accumulated for multiple controls of heterochromatin formation by SUMO. As already mentioned, it was initially reported that SUMOylation of core histones can be associated with heterochromatin protein HP1γ recruitment [264] and that histone SUMOylation is enriched at telomeres in budding yeast [265]. SUMOylated TFs can also attract HP1 proteins. For example, SUMOylated Sp3 bound to DNA is associated with transcriptional repression leading to compacted repressive chromatin presenting features of heterochromatin. This involves the Mi-2 component of the ATP-dependent chromatin remodeler NuRD, the heterochromatin proteins HP1α, -β and –γ and the HMTs SETDB1 and SUV4-20 concomitantly with the establishment of histone modifications associated with repressed genes (H3K9me3 and H3K20me3) [285]. These observations are pertinent in living organisms, as demonstrated in transgenic mice where transcriptional repression in spermatocyte-specific and neuronal genes was also associated with higher DNA methylation [286]. Interestingly, a non-SUMOylatable mutant of HP1 in *Schizosaccharomyces pombe* was found less efficiently recruited in heterochromatin, which correlated with decreased HP1-mediated repression of transcription [287]. Moreover, studies in murine cells showed that SUMOylation of the heterochromatin protein HP1α regulates its de novo localization to pericentromeric chromatin [288]. It was later shown that the H3K9 methylase Suv39h1 acts as the SUMO E3 in this process, the latter function lying in a peptidic region different from the KMT catalytic domain [289]. Moreover, the SENP7 isopeptidase was found to interact with HP1α and to be necessary for restricting HP1α mobility at pericentromeric domains, which is essential to ensure heterochromatin stability and subsequent proper chromosome segregation during mitosis [290,291]. Whether this applies to other heterochromatin domains still requires investigation. It is however of note that SENP7 promotes chromatin relaxation for homologous recombination DNA repair [292], indicating multiple possible roles for it in control of chromatin configuration.

Another noteworthy situation in mammalian cells is that of endogenous retroviruses. ERVs are vestiges of the long history of coevolution with retroposons that have shaped the genome. ERV proviruses are associated with H3K9me3 and their transcriptional silencing through heterochromatinization is important to avoid mutagenic events that would result from their retroposition [293]. A genome-wide siRNA screen [294] and a CRISPR knock-out screen [295] identified SUMOylation factors as critical repressors of ERVs in ESCs. Moreover, SUMO was separately found enriched at H3K9me3-marked regions in a genome-wide ChIP-seq analysis conducted in mouse ESCs, including at ERV loci, and its depletion led to global reduction of H3K9 trimethylation and ERV derepression [137]. SUMOylation of the factor Trim28/TIF1b/KAP1 is crucial in the heterochromatinization process [294]. Importantly, Trim28/TIF1b/KAP1 carries a SUMO E3 activity permitting its autoSUMOylation [296]. This event enhances the recruitment of Trim28/TIF1β/KAP1 to proviral DNA, which in turn facilitates the modification of proviral chromatin with the repressive H3K9me3 marks. Precise targeting of H3K9me3 deposition at ERVs is an important issue and, if H3K9 HMTs and HP1 proteins bear some DNA-binding ability, they can however not recognize specific DNA motifs. This specificity must therefore be brought by other factors, namely members of the family of the Krüppel-associated (KRAB)-containing zinc finger proteins (KRAB-ZNFs). In this scenario, KRAB-ZNFs first bind to their TFBSs found at ERVs loci. Then, they recruit the universal repressor Trim28/TIF1β/KAP1, which, after autoSUMOylation, recruits the HMT SetDB1 via SUMO/SIM interactions, permitting deposition of the H3K9me3 mark [293].

## 7. Conclusions

In the recent years, large-scale proteomics-, modifomics- and genomics studies have considerably improved our view of the role of SUMO in transcriptional regulation and unveiled novel layers of mechanistical complexity. Coupled to functional studies, they have shown that the three types of transcription (Pol I, -II and -III) are concerned via SUMOylation of numerous molecular actors. These range from transcription factors and transcription co-regulators up to the transcriptional machineries themselves, as well as to many chromatin components and their organizers/remodelers. These studies also have established the existence of a complex dialog between SUMOylation and other post-translational modifications operating in the cell through molecular mechanisms that can be cooperative or antagonistic, depending on the situation. Numerous challenges however still remain ahead of us before featuring a precise picture of the many roles of SUMO in transcription. In particular, not only the complete repertoire of the post-translational modifications of all transcriptional actors will have to be established in a cell signaling- and/or development-dependent manner, but these data will also have to be integrated in scenarii taking into account the structure of the genome and its diverse transcription regulatory elements without forgetting the dynamics of its tridimensional organization. It will also be necessary to understand how SUMOylation events are coordinated in, and between, the supramolecular complexes involved in transcription and how this molecularly serves transcriptional control. Achieving this goal will undoubtfully require novel technological developments and investigation methodologies.

## Figures and Tables

**Figure 1 molecules-26-00828-f001:**
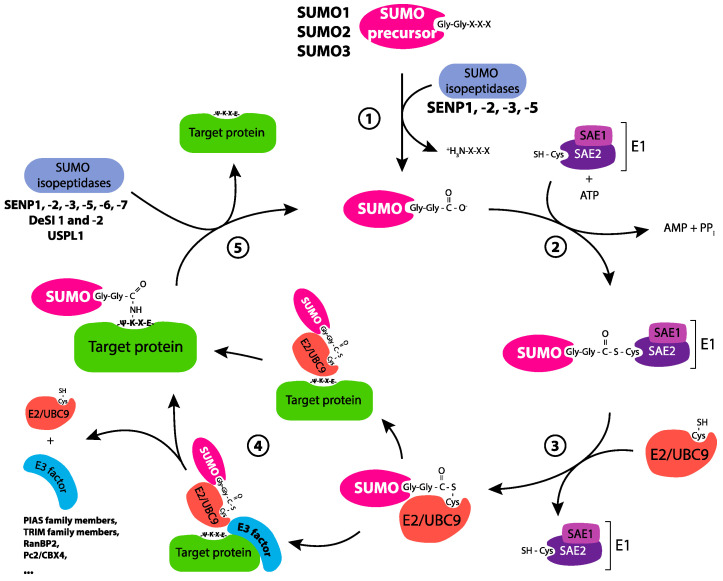
The SUMOylation cycle. The C-terminal extensions of SUMO-1, -2 and -3 precursors are removed by specific peptidases from the SENP family (SENP1, -2, -3 and -5, which also harbor isopeptidasic activities) to expose a C-terminal Gly-Gly motif essential for SUMO activation and, then, conjugation to protein substrates (1). SUMO is activated in an ATP-dependent manner via transfer to a reactive cysteine of the SAE2 subunit of the heterodimeric E1 SUMO-activating enzyme (2). This occurs in two steps. First, the SUMO GG motif is adenylated by the E1 enzyme. Then, full activation is obtained by transfer of the adenylated SUMO intermediate to the SAE2 reactive cysteine. Activated SUMO is then transferred to the reactive cysteine of the E2 SUMO-conjugating enzyme also called Ubc9 (3) and covalently coupled to the ε-NH_2_ group of target lysines of protein substrates owing to an isopeptide bond. This can be achieved directly or with the help of SUMO E3 factors assisting the SUMO E2 enzyme (4) and often, but not always, occurs at a ψKxE consensus motif (where ψ corresponds to a bulky hydrophobic amino acid and x to any amino acid). Some SUMO E3 factors are indicated but others probably remain to be discovered. Certain enzymes display an E4 elongase activity facilitating the formation of polySUMO chains (not represented). SUMO can be removed from its substrates and SUMO chains can be depolymerized by proteases showing SUMO isopeptidasic activity (5). These include the members of the SENP family as well as a few other enzymes such as DeSI-1 and -2 and USPL1.

**Figure 2 molecules-26-00828-f002:**
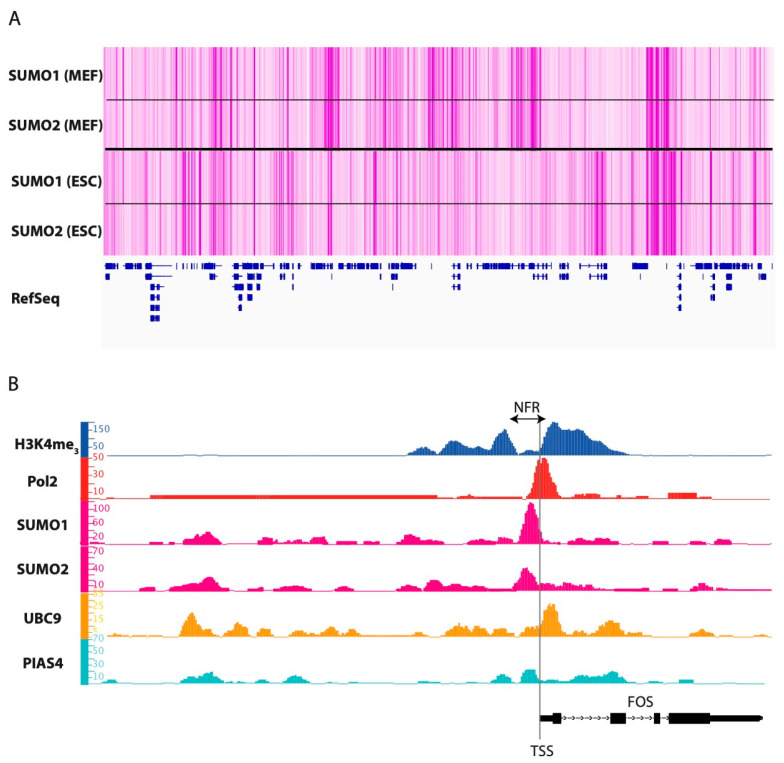
SUMO distribution on chromatin. (**A**) SUMO distribution on chromatin. SUMO is found at many discrete sites in ChIP-seq experiments, as illustrated here for a portion of mouse chromosome 9 (GSE99009). However, whereas signal distribution can be very similar between SUMO-1 and SUMO-2/3 in the same cellular context, it can vary to a great extent from one to another cell type, as shown here between mouse embryo fibroblasts (MEF) and mouse embryonic stem cells (ESC) [137]. SUMOylation patterns can also change in response to different signaling cues (see Text). (**B**) SUMO at a transcriptionally active gene. The c-Fos gene is taken as an illustrative example using ChIP-Seq data obtained in human primary fibroblasts (GSE42213) [175]. The transcriptional start site (TSS) lies in a nucleosome free region (NFR) around which Histone 3 is marked by trimethylation of its lysine 4, H3K4me3 usually being associated with transcriptional activity. High Pol II signal just downstream of the TSS is indicative of c-Fos as a paused gene (see Text for details). SUMO-1 and SUMO-2 signals peak in the NFR just upstream of the TSS but SUMO signals are also observed at discrete locations all over the locus. The SUMO E2 (Ubc9) and a SUMO E3 (PIAS4) are also found all over the locus. However, colocalization of SUMO, Ubc9 and PIAS4 is not strict, as found at many other places on the genome.

**Figure 3 molecules-26-00828-f003:**
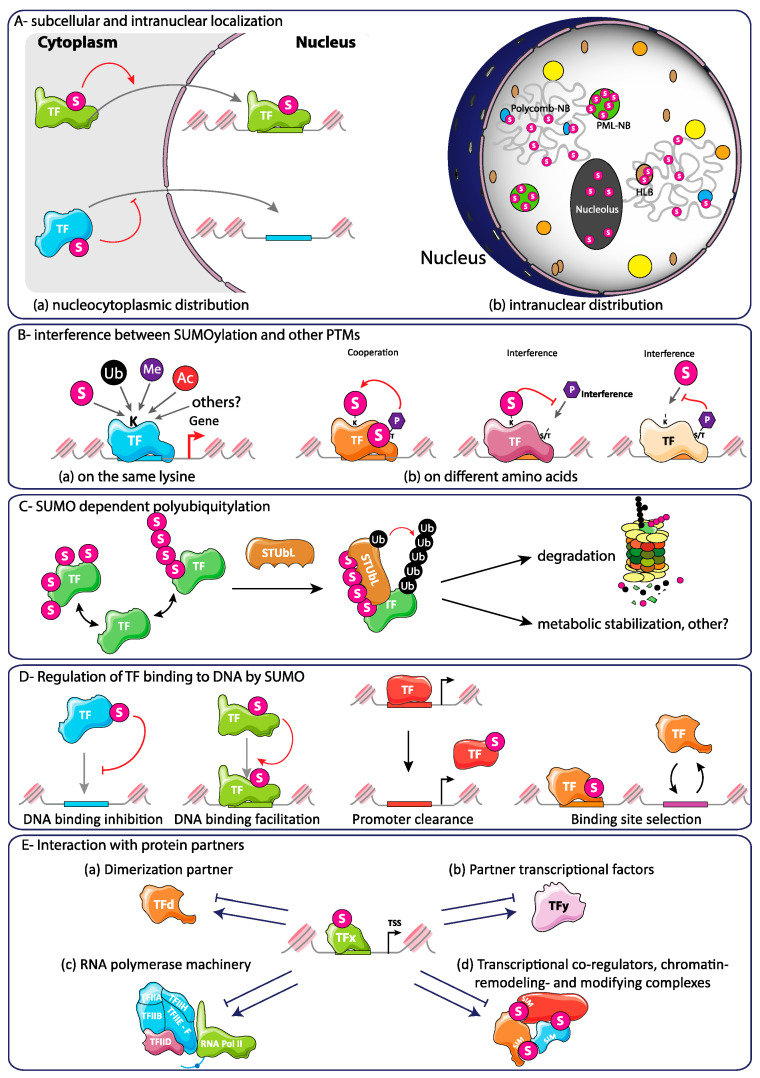
Alteration of transcription factor properties by SUMOylation. SUMOylation can alter various properties of TFs at various stages of their actions and/or their lifetime and in a finely tuned manner. (**A**) Subcellular and intranuclear localization. SUMOylation can control nucleo-cytoplasmic shuttling of certain TFs (**a**), as well as their addressing to the various intranuclear subcompartments, including chromatin, PML bodies, Polycomb bodies, histone locus bodies, nucleoles and Cajal bodies (**b**). (**B**) Interferences with other post-translational modifications. SUMOylation can cross-talk with other post-transcriptional modifications that control transcriptional activity and/or protein fate. For example, SUMOylation, ubiquitylation and acetylation can compete for the modification of the same lysines (**a**). SUMOylation can also be dependent upon certain phosphorylations and SUMOylation and phosphorylations at nearby or distant sites can exert cooperative or antagonistic effects on TF activity (**b**). (**C**) SUMO-dependent polyubiquitylation. PolySUMOylated or multimonoSUMoylated TFs can interact with StUbLs, leading to their polyubiquitylation. This can lead to subsequent addressing to the proteasome but also, sometimes, to protein stabilization. (**D**) Binding and release from DNA/chromatin. SUMOylation of TFs can alter binding to and clearance from their TFBSs in various ways. SUMOylation of chromatin-bound TFs can also participate in TFBS selection via releasing TFs from less affine TBFSs. (**E**) Interaction with partners. SUMOylation can also alter positively or negatively TF interactions with their protein partners with varied effects on transcriptional outcomes. These include: homo or heterodimerization in the case of dimeric TFs (**a**), interactions with other TFs (**b**) transcriptional machineries (**c**) and various kinds of transcriptional coregulators or chromatin-modeling and modifying complexes (**d**).

**Figure 4 molecules-26-00828-f004:**
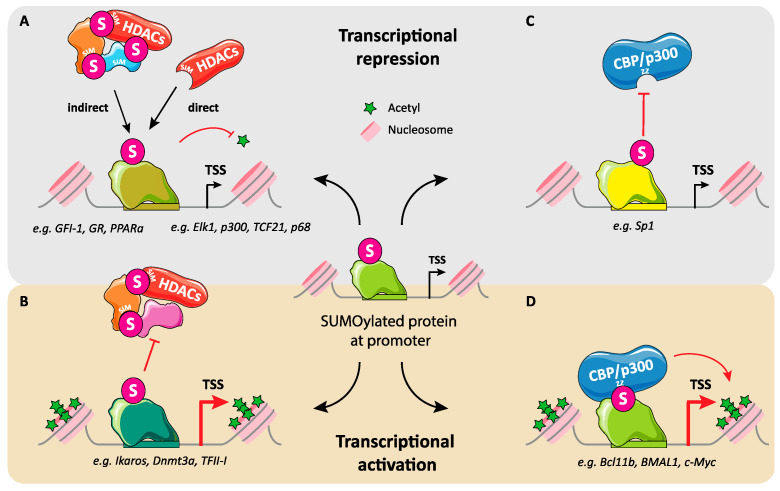
Interplay between SUMO, HDACs and HATs. Histone acetyltransferases and histone deacetylases are key co-transcriptional regulators. Their action can be favored or disfavored by SUMO conjugated to either transcription factors, other cotranscriptional regulators or even themselves. Several generic situations are presented here. They are non-exhaustive. Examples of SUMOylated TF are given for each situation (**A**) HDACs recruitment by SUMOylated TFs. This can be achieved directly or indirectly through SIM-SUMO interactions, leading to chromatin deacetylation and, thereby, transcriptional repression. (**B**) Inhibition of HDACs recruitment by SUMOylated TFs. This permits higher levels of chromatin acetylation and favors transcription. (**C**) Inhibition of HAT recruitment by SUMOylated TFs. p300/CBP are archetypical HATs. Inhibition of their recruitment at promoters by SUMOylated TFs diminishes histone acetylation, which reduces transcriptional activity. (**D**) Recruitment of HATs by SUMOylated TFs. In certain cases, SUMOylated TFs can facilitate HAT recruitment at gene regulatory elements. As HAT and HDACs associate with regulatory factors and other transcriptional cofactors into multi-subunit complexes, it is likely that the regulation of their activity involves group SUMOylation (see Text for details). The same remark holds true for other chromatin-modifying complexes such as KMTs and possibly others whose regulation by SUMO has not been studied yet.

**Figure 5 molecules-26-00828-f005:**
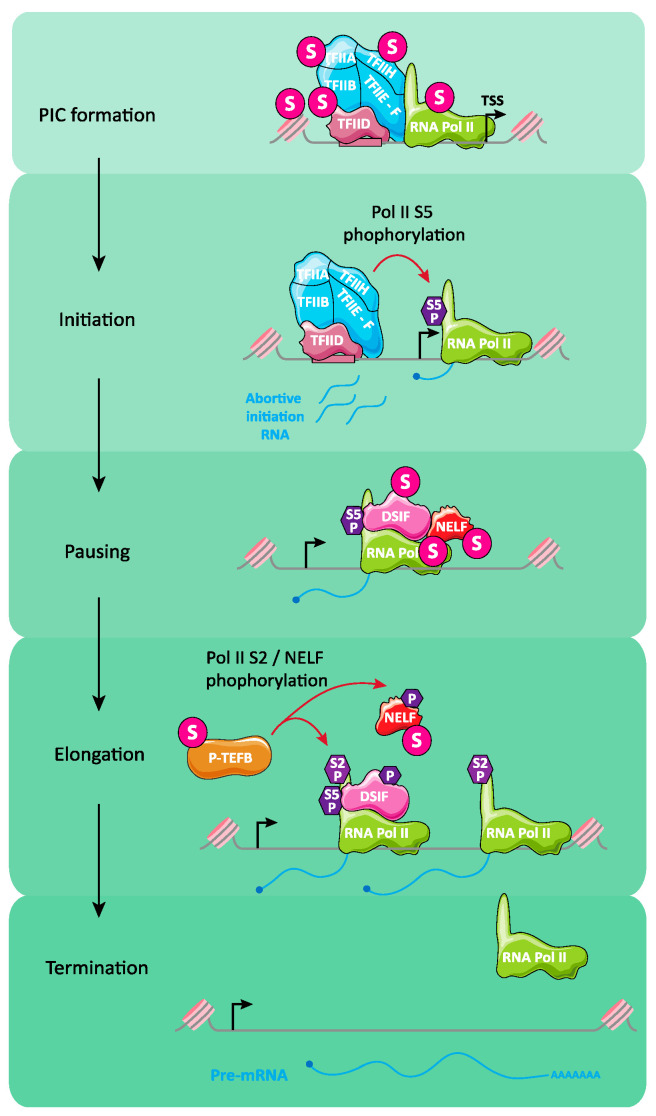
SUMO in transcription initiation and elongation by Pol II. PIC formation: the pre-initiation complex forms just upstream of the transcriptional start site in a nucleosome-free domain, provided that chromatin is competent for transcription due to the presence of transcription factors bound to their TFBSs, appropriate chromatin access and modifications, etc. (not represented). The general transcription factor TFIID, which binds to DNA, is crucial in this process, as in permits the recruitment of other GTFs as well as Pol II. Pausing of Pol II downstream of the TSS is permitted by the NELF and DSIF multisubunit factors. Release of Pol II on the gene is then permitted by the P-TEFb dimeric factor. Termination of transcription occurs downstream the polyadenylation site. Various subunits of GTFs, Pol II complex, NELF, DSIF can undergo SUMOylation, which alters their activity and, thereby, transcriptional activity (see Text for details on SUMO actions). To which extent and how group SUMOylation is involved in the regulation of transcription initiation and elongation, possibly in a phase separation-dependent manner, is not known. The implication of SUMO in transcriptional termination has not been studied yet.

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
