# Peer review of "SUMO and Transcriptional Regulation: The Lessons of Large-Scale Proteomic, Modifomic and Genomic Studies"

_molecules, 2021, doi:10.3390/molecules26040828_

Round 1
Reviewer 1 Report
The manuscript “SUMO and Transcriptional Regulation: the lessons of large-scale proteomic, modifomic and genomic studies” by M. Boulanger, M. Chakraborty, D. Tempé, M. Piechaczyk and G. Bossis is a very comprehensive and timely review devoted to the different aspects of functioning and activity of SUMO family proteins with an emphasis on SUMO participation in transcription-related processes. I agree with the authors that large amount of data published on SUMO (Small Ubiquitin-like Modifier) proteins to date requires systematization and understanding. At the same time, when considering issues related to transcription, a broader view of the processes in which SUMO proteins are involved is needed. In my opinion, the authors have successfully coped with the task, and the review can be recommended for publication after a small revision.
Remarks for revision:
- I propose to delete Tables 1 and 2, because the current content of the tables is very simple and does not require a tabular presentation. Moreover, content of tables 1 and 2 is already present in the text. If necessary, the content of the tables can be presented in the text more precisely as a list of enumerations.
- Citations like (see [6,7,9,10]) should be presented in conventional style [6,7,9,10] (throughout the text).
- For the SUMO-activating enzymes, two types of abbreviation are used SUMO E1 and E1. It is better to use the only type of abbreviation , and the style E1 ( or E1 enzyme, or E1 SUMO-activating enzyme) is preferable to avoid confusion between complexes/conjugates of SUMO with E1 and SUMO E1 as the E1 SUMO-activating enzyme. The same is with E2 and E3.
- Line 101. Change …” from the E2 enzyme with, or without, the help of members…” to “ from the E2 enzyme with (or without) the help of members”
- Legend to Figure 1 is very similar (in many places) to the sentences in text, thus duplicating them. Please, correct it where possible.
Sentence “SUMO-1 is usually less abundant than SUMO-2, which is itself less abundant than SUMO-3” is very similar to a sentence in the text (lines 81-83). I propose to delete it from the figure legend. Additionally, “The pool of free SUMO-1 is usually very limited whereas those of SUMO-2 and -3 can be larger, though SUMO is mostly found conjugated to its substrates in living cells” should be transferred into the main text.
There is a strange symbol between “the” and “NH2 group” in the figure legend in the phrase “coupled to the NH2 group of target lysines” (at least in the pdf file). The same symbol is used in the fragment “a KxE consensus motif (where correspond…” instead of Ψ.
- Line 144. “They also include cross-talks between these functions…” They (?) – not clear who are “they”.
- Line 280. Please change ([120,121]. Also see [122]) to [120-122].
- Figure 4. A set of exactly identical partners is shown in panels C and D. Therefore, it is not clear, why their interaction is blocked in panel C, but in panel D, on the contrary, it is allowed.
- Line 906 correct “subtructures”
Author Response
We thank this reviewer for his insightful comments, which allowed to improve our review
- 1/ Table 1 and 2 have been deleted and part of their information included in the main text.
- 2/ The concerned citations are presented as requested by Reviewer 1.
- 3/ We have adopted a single format for SUMO E1, -E2 and -E3s throughout the text.
- 4/ line 101: corrected to facilitate understanding.
- 5/ Legend to Figure 1 has been shorten. As suggested by Reviewer 1, part of the information was transferred to the main text. The Greek symbols may have been converted in "strange symbols" in the original pdf copy submitted to the reviewers. They have been verified in the final Word version of the main text.
- 6/ Clarification made.
- 7/ Reference modified
- 8/ Figure 4 was adapted as requested
- 9/ Correction made.
Reviewer 2 Report
In their manuscript, Boulanger et al review recent advances in our understanding of the diverse functions of Sumo in regulation of transcription. This is a well-written, exhaustive review that provides an excellent overview of the current status of the field. I am sure it will serve as a sublime reference for many researchers in this field and gather a large number of citations. The figures are well designed and informative. I only have relatively minor recommendations to help the authors improve their manuscript:
- One general comment is that there is a certain degree of redundancy in the text. For instance: “Finally, SUMOylation of TFs themselves can be a complex phenomenon involving multiple additions of SUMO moieties, or the formation of polySUMO chains, whose outcomes may also be diverse with the intervention of StUbLs such as RNF4 or Arkadia directing them to either proteasomal degradation or, on the contrary, stabilization via the addition of polyubiquitin chains”. There is considerable overlap between this sentence and an earlier description StUbLs. The authors may want to scan the document for such redundancies and remove/merge them as much as possible to streamline and sharpen the text.
- The authors can expand a little on limitations of ChIP-seq analysis of Sumo. These experiments provide snapshots of steady-state Sumo localization, but high levels of Sumo occupancy do not necessarily correlate with where the action is. This has been explored for certain TFs like budding yeast Rap1, where it was found that TF residence time appears to be the transcriptionally most relevant pool (Lickwar et al, Nature 2012, PMID: 22498630).
- The authors mention a rather large number of TFs that can be modified by different Sumo isoforms. In some cases this impairs readability of the text (such as e.g. paragraph between Line 808-833, but there are many such instances). The authors should consider reducing the burden of all these protein names in the text, and rather focus on a single well-explained example, while referring the reader for additional examples to a compiled table (and/or figures) that summarizes the other modified proteins, and which very briefly explains the effect of the modification on transcription with references to the relevant studies. Such a table would also be a useful tool for researchers to quickly look up what is known about sumoylation of their protein of interest.
- Table 1 and 2 only offer limited information. Perhaps the authors can expand these tables to include more information, e.g. Table 2 could include the relevant Sumo substrates and references to the relevant studies. Table 1 could be converted into a figure with different panels showing graphically the possible effects of Sumo (this is partially done in Fig. 3).
- The authors should expand on spatiotemporal regulation of transcription by Sumo in the nucleus. There are some interesting studies on this topic, such as by Texari et al (Mol Cell 51(6) 807-818).
- Line 501: The section regarding the effect of Sumo on RNAPIII activity in budding yeast might be a bit difficult to follow for the general audience, primarily because the literature is somewhat complex and at first glance contradictory; i.e. Sumo has been shown to be able to both activate and inactivate RNAPIII activity. The authors may want to restructure this section to more clearly contrast these differences. For example, it could be mentioned that under normal growth conditions, Sumo likely promotes the activity of RNAPIII, because depletion of Ubc9 strongly reduces tRNA synthesis (Chymkowitch et al, 2017). This indicates that in unstressed cells the dominant function of Ubc9-Sumo is to promote RNAPIII activity. However, as the authors correctly point out, Sumo is likely to have inhibitory effects on RNAPIII activity under specific stress conditions (which e.g. might lead to desumoylation of activatory sites in the RNAPIII machinery and increased sumoylation of inhibitory sites)
Minor issues:
-Phase separation (line 347): Please provide some background to introduce this process to non-expert readers
-Line 773: “This is also that…” Cryptic description, please reformulate
Author Response
We thank this reviewers for his/her insightful comments, which allowed to improve our review.
-1/ The document has been scanned as requested by Reviewer 2 to limit redundancies.
- 2/ This is an important notion that has been introduced in the "General lessons of genome-wide studies of chromatin SUMOylation" section.
- 3/ We have simplified the text wherever possible.
- 4/ Tables 1 and 2 were canceled has suggested by Reviewer 1 and part of its information included in the main text.
- 5/ The main message of the Texari et al.'s paper is now included in the review.
- 6/ This section has been improved to make it clearer to the general readership.
- 7/ General information on phase separation is now provided to make this notion clearer to the general readership.
- 8/ Correction made